# Confocal Endomicroscopy of Neuromuscular Junctions Stained with Physiologically Inert Protein Fragments of Tetanus Toxin

**DOI:** 10.3390/biom11101499

**Published:** 2021-10-12

**Authors:** Cornelia Roesl, Elizabeth R. Evans, Kosala N. Dissanayake, Veronika Boczonadi, Ross A. Jones, Graeme R. Jordan, Leire Ledahawsky, Guy C. C. Allen, Molly Scott, Alanna Thomson, Thomas M. Wishart, David I. Hughes, Richard J. Mead, Clifford C. Shone, Clarke R. Slater, Thomas H. Gillingwater, Paul A. Skehel, Richard R. Ribchester

**Affiliations:** 1Centre for Discovery Brain Sciences and the Euan MacDonald Centre for Motor Neurone Disease Research, University of Edinburgh, George Square, Edinburgh EH8 9XD, UK; cornelia.roesl@lifearc.org (C.R.); Kosala.Dissanayake@ed.ac.uk (K.N.D.); Ross.Jones@ed.ac.uk (R.A.J.); graemerjordan@gmail.com (G.R.J.); Leire.Ledahawsky@ed.ac.uk (L.L.); gallen96@hotmail.com (G.C.C.A.); scottmk212@gmail.com (M.S.); alanna.thomson@hotmail.co.uk (A.T.); T.Gillingwater@ed.ac.uk (T.H.G.); 2Public Health England, National Infection Service, Porton Down, Salisbury SP4 0JG, UK; Liz.Evans@phe.gov.uk (E.R.E.); research.porton@phe.gov.uk (C.C.S.); 3Applied Neuromuscular Junction Facility, Bio-Imaging Unit, Biosciences Institute, University of Newcastle-upon-Tyne, Framlington Place, Newcastle-upon-Tyne NE2 4HH, UK; veronika.boczonadi@newcastle.ac.uk (V.B.); c.r.slater@newcastle.ac.uk (C.R.S.); 4The Roslin Institute, Royal (Dick) School of Veterinary Studies, College of Medicine and Veterinary Medicine, University of Edinburgh, Easter Bush, Edinburgh EH25 9RG, UK; T.M.Wishart@ed.ac.uk; 5Spinal Cord Research Group, Institute of Neuroscience and Psychology, University of Glasgow, Glasgow G12 8QQ, UK; David.I.Hughes@glasgow.ac.uk; 6Sheffield Institute for Translational Neuroscience (SITraN), University of Sheffield, Glossop Road, Sheffield S10 2HQ, UK; r.j.mead@sheffield.ac.uk

**Keywords:** neuromuscular junction, tetanus toxin, imaging

## Abstract

Live imaging of neuromuscular junctions (NMJs) in situ has been constrained by the suitability of ligands for inert vital staining of motor nerve terminals. Here, we constructed several truncated derivatives of the tetanus toxin C-fragment (TetC) fused with Emerald Fluorescent Protein (emGFP). Four constructs, namely full length emGFP-TetC (emGFP-865:TetC) or truncations comprising amino acids 1066–1315 (emGFP-1066:TetC), 1093–1315 (emGFP-1093:TetC) and 1109–1315 (emGFP-1109:TetC), produced selective, high-contrast staining of motor nerve terminals in rodent or human muscle explants. Isometric tension and intracellular recordings of endplate potentials from mouse muscles indicated that neither full-length nor truncated emGFP-TetC constructs significantly impaired NMJ function or transmission. Motor nerve terminals stained with emGFP-TetC constructs were readily visualised in situ or in isolated preparations using fibre-optic confocal endomicroscopy (CEM). emGFP-TetC derivatives and CEM also visualised regenerated NMJs. Dual-waveband CEM imaging of preparations co-stained with fluorescent emGFP-TetC constructs and Alexa647-α-bungarotoxin resolved innervated from denervated NMJs in axotomized *Wld^S^* mouse muscle and degenerating NMJs in transgenic SOD1G93A mouse muscle. Our findings highlight the region of the TetC fragment required for selective binding and visualisation of motor nerve terminals and show that fluorescent derivatives of TetC are suitable for in situ morphological and physiological characterisation of healthy, injured and diseased NMJs.

## 1. Introduction

Differential diagnosis of neuromuscular abnormalities or motor neuron diseases such as amyotrophic lateral sclerosis (ALS) is commonly based on combinations of biomarker readings, neurophysiological investigation, pathology in muscle biopsies and clinical experience with reference to agreed international protocols [1,2,3,4,5]. Non-invasive or minimally invasive imaging methods using, for example, computerised tomography (CT) or magnetic resonance imaging (MRI) can also provide insight into gross neuromuscular pathology [6,7,8,9]. However, early signs of motor neuron pathology or axonal injury include disruption of motor nerve terminals and endplates at neuromuscular junctions (NMJs) and to visualise this pathology in situ requires micrometre image resolution, which cannot currently be achieved using techniques such as CT or MRI [10,11,12,13,14,15,16]. 

Confocal endomicroscopy (CEM) utilises laser scanning and fluorescence capture via hundreds of optical filaments contained in narrow (0.5–1.5 mm) probes. This technology has enabled real time (up to 12 video frames per second), direct imaging of cells either selectively expressing fluorescent proteins or stained with exogenous ligands conjugated to specific fluorochromes [17,18,19]. We showed previously that CEM visualises motor nerve terminals in mice expressing Yellow Fluorescent Protein (YFP) selectively in motor neurons [20]. Innervated or denervated NMJs could also be distinguished by combining CEM with conventional electromyography in these mice [21]. However, wider application of CEM for research, or veterinary or clinical diagnostic applications, would require non-toxic, high-contrast fluorescent ligands for axons and NMJs that can be administered exogenously, for example by local subcutaneous or intramuscular injection. We confirmed previously that 4-di-2-Asp, a fluorescent styryl dye that passively stains motor nerve terminals [22,23], and a green fluorescent protein (GFP)-tagged construct of a botulinum toxin heavy chain that also binds to NMJs [24,25] both readily enabled visualisation of motor nerve terminals using conventional wide-field or laser scanning confocal microscopy (LSCM). However, the intensity and contrast of the fluorescence of these molecules were insufficient for routine visualisation of NMJs using fibre-optic CEM, whose sensitivity is less than that of conventional LCSM systems or wide-field fluorescence microscopes [21].

Motor nerve terminals contain ganglioside and peptide binding sites that selectively adhere fluorescent conjugates of the tetanus toxin heavy chain (comprising non-toxic B and C fragments), from which the toxic light chain (A fragment) has been removed [26,27,28,29]. Remarkably, truncations of TetC from the N-terminus were shown previously to improve binding to ganglioside receptors [30]. In the present study we therefore investigated the potential for staining and CEM visualisation of motor nerve terminals using full length and truncated derivatives of the C-fragment (TetC), fused at the N-terminus to Emerald Fluorescent Protein (emGFP), a brighter variant of Green Fluorescent Protein [31], that retained the nerve terminal receptor binding domain [32,33,34]. We tested the suitability of these potential ligands for vital staining of motor nerve terminals and their visualisation with single or dual-waveband CEM.

We found that fusion proteins containing 206 or more amino acids from the C-terminus of TetC produced high-contrast, selective staining of motor nerve terminals. Electrophysiological analysis showed that emGFP-TetC constructs were physiologically inert since they did not acutely alter neuromuscular synaptic transmission or function. Nerve terminal staining was readily visualised using CEM optical fibre probes when these were applied to skeletal muscles of either healthy mice, or to slowly-degenerating axotomized terminals in *Wld^S^* mutant mice or in the *SOD1G93A* mouse model of motor neuron disease. Our findings show that one-step vital staining of nerve terminals with truncated TetC fragments and labelled α-bungarotoxin (BTX), combined with dual-waveband CEM imaging, comprise a potentially powerful combination that could facilitate analysis of neuromuscular synaptic form and function at normal, damaged or diseased NMJs. 

## 2. Materials and Methods

### 2.1. Ethics, Animals and Tissues

Isolated nerve-muscle preparations were made from adult (1–8 month old) stock mice of C57Bl6, CD01, *Wld^S^*, *thy1.2YFP16-Wld^S^*, or *SOD1G93A* strains bred and maintained in University of Edinburgh, University of Sheffield or University of Glasgow animal care facilities under conditions approved by local ethical committees, closely monitored by appointed Veterinary Officers, and regularly inspected under licence by the UK Home Office.

With local ethical committee approval and under Home Office Licences, some mice were anaesthetised with isoflurane and volumes of saline (up to 100 µL) containing physiologically-inert, Emerald Fluorescent Protein (emGFP)-tagged tetanus toxin C fragments (emGFP-TetC; see below) were injected subcutaneously in one hind limb. The animals were sacrificed 20–30 min later. In other experiments, mice were injected with 5 mg/kg carprofen and/or 0.05–0.3 mg/kg buprenorphine, anaesthetised with isoflurane and the sciatic nerve in the thigh was then exposed on one side through a small skin incision. For study of slow nerve-terminal degeneration in *Wld^S^* mice, the nerve was cut with micro-scissors; and for study of axonal and nerve terminal regeneration, in wild-type mice, the nerve was crushed between the tips of fine forceps. The wounds were closed with 6/0 silk suture or wound clips before returning mice to home cages for 5–21 days. Mice were inspected daily for signs of discomfort by veterinary surgeons or other qualified animal care staff.

For terminal investigations, animals were sacrificed by anaesthetic overdose and cervical dislocation, in compliance with UK Home Office Schedule 1. *Extensor digitorum longus* (EDL), *soleus* (SOL), *flexor digitorum brevis* (FDB) or deep lumbrical (DL) muscles in the hind limbs, *epitrochleoanconeus* (ETA) in the forelimb, or *triangularis sterni* (TS) muscles in the thorax were rapidly dissected and immersed in mammalian physiological saline (MPS) of composition (mM): NaCl (158), KCl (5), CaCl_2_ (2), MgCl_2_ (1), glucose (5), HEPES (5), adjusted to pH 7.2–7.4 with HCl or NaOH. Solutions were bubbled with air for at least 20 min.

### 2.2. Human Tissue: Ethics and Sampling

Anonymised human tissue was obtained in accordance with the guidelines of the Declaration of Helsinki and approved by the Institutional Review Board of the NHS Lothian Ethics Committee (REC 2002/1/22; 2002/R/OST/02) and NRS BioResource (15/ES/0094, SR719, SR769; 15/SS/0182, SR589) and with informed consent. Muscle samples were obtained as described previously [35] from surgical patients undergoing lower limb amputation for complications of peripheral vascular disease, typically either critical ischemia in a non-salvageable limb, or failure of previous vascular reconstruction. Samples of the *soleus* muscle were dissected in the operating theatre from the discarded distal limb immediately after its surgical amputation. Tissue was harvested from the proximal end, close to the line of surgical incision and away from areas of necrosis or infection. Small blocks of tissue, containing full-length muscle fibres from origin to insertion (approx. 2 cm in length) were removed from each of the muscles selected and immersed in vials containing 10–20 mL of standard 0.2 M phosphate-buffered saline (PBS). The specimens were then dispatched to the laboratory, where small bundles of muscle fibres in the region of innervation were dissected in MPS. 

### 2.3. GFP-TetC Constructs

A full-length synthetic tetanus toxin C-fragment gene was synthesised in similar fashion to that described previously [36]. For tagging with Green Fluorescent Protein (GFP), a fully synthetic genetic sequence was prepared comprising the GFP gene from *Aequorea Victoria* and the Hc region sequence from the *Clostridium tetani* neurotoxin (TeNT-Hc) was synthesised using a codon bias optimised for expression in *Escherichia coli*. Restriction sites *BamHI* and *XbaI* were incorporated at the 5′ and 3′ ends, respectively, to allow subcloning into expression vectors. GFP-TeNT-Hc-encoding fragments were subcloned into the vector pMAL-c2x (New England Biolabs, Ipswich, MA, USA) and expressed as maltose binding protein (MBP) fusions in the host strain BL21 (Novagen, Darmstadt, Germany). A single colony was inoculated into 200 mL of Terrific Broth (24 g/L yeast extract, 12 g/L tryptone, 9.4 g/L KH_2_PO_4_, 2.2 g/L K_2_HPO_4_, pH 7.2) supplemented with 100 µg/mL ampicillin and 0.5% (*w*/*v*) glucose and grown overnight at 30 °C in a shaking incubator. Cultures were diluted 1:6 with fresh medium and grown to an OD600 of approximately 2.0 at 30 °C. IPTG was added (500 μM) and cultures grown for 2 h at 25 °C. Cell pellets were resuspended in 20 mM Tris-HCl pH 7.4, 500 mM NaCl, 1 mM EDTA (binding buffer) and lysed by sonication. Cell debris was removed by centrifugation at 27,000× *g* and the supernatant fluid applied to a 5 mL amylose resin (NEB) column equilibrated in the binding buffer. After washing with binding buffer, the fusion protein was eluted with binding buffer supplemented with 10 mM maltose. MBP was removed from fusion proteins by treatment with factor Xa protease (1 U/mg) at 20 °C for 18 h in the elution buffer. GFP-TetC proteins were purified by anion exchange (Mono-Q) chromatography and then dialysed against 20 mM HEPES pH 7.4 containing 200 mM NaCl.

### 2.4. Truncated TetC Constructs

Overlapping, complementary forward and reverse primers were generated using the sequence for tetanus toxin heavy chain found on Uniprot (entry: PO4958) or NCBI (AM412776). The tetanus C-fragment sequence was inserted into a pET28 vector downstream of eGFP and a HIS6-tag. For the replacement of eGPF by emGFP, the emGFP sequence was ordered as a gBlock from Integrated DNA Technologies (Leuven, Belgium) with an *NdeI* site at the 5′ end and a *BamHI* site at the 3′ end. The sequence was amplified by hi-fidelity PCR and the purified PCR product was digested with *NdeI* and *BamHI* in NEB4 buffer before purification using the Zyppy Clean (Zymo Research, Cambridge, UK) and concentrator kit according to manufacturer’s instructions. The insert was ligated into the *NdeI* and *BamHI* cut pET28-eGFP-TetC vector. Ligation was done in a 1:3 ratio and incubated overnight at room temperature. Constructs were transformed into Invitrogen Top10 competent *E. coli* cells and plasmid extracted using the Qiagen plasmid mini prep kit according to manufacturer’s instructions. All constructs were confirmed by Sanger sequencing. Truncations were designed for regions of interest, according to the crystal structure of the heavy-chain from EMBL-EBI [37] and effected by restriction free In-Fusion cloning generating the indicated deletions of the N-terminal region of TetC. Primer sequences will be provided upon request to the appropriate corresponding author (P.A.S.).

### 2.5. DNA Extraction from Bacteria

Top10 competent *E. coli* were grown in 5 mL or 100 mL and the Qiagen Mini or Midi prep kit, respectively, were used according to manufacturer’s instruction. Purified DNA was then used in either further cloning experiments or protein expression.

### 2.6. Bacterial Expression and Protein Purification

Rosetta-DE3-pLys cells were electroporated with 10 ng of expression plasmid DNA and plated on Kanamycin + Chloramphenicol plates and grown overnight at 37 °C. Single colonies were used to inoculate LB broth and grown to an OD_600_ of ~0.7 before adding isopropyl b-D-thiogalactopyranoside (IPTG) to 1.5 mM and 2% ethanol and incubated at room temperature overnight. Cultures were harvested by centrifugation (6000 rpm, 4 °C, 15 min) and the bacterial pellet washed once in PBS. The pellet was then resuspended in lysis buffer (10 mM HEPES, pH 7.9, 500 mM NaCl, EDTA-free protease inhibitor Complete (Roche) and the cells sonicated (six times, 30 s) before a final centrifugation (13,000 rpm, 30 min) to remove bacterial debris. The supernatant was purified on a Ni^2+^NTA column (Qiagen, Venlo, The Netherlands) and eluted using an Imidazole gradient from 50 mM to 350 mM. Eluted fractions were pooled and dialyzed against PBS using a Vivaspin4 spin column. Protein was stored at −80 °C for long term storage or at 4 °C for short term storage.

### 2.7. Fluorescence Measurements of TetC Conjugates

Fluorescence of emGFP and eGFP constructs were compared fluorometrically. Standard curves ranging from 0.05–1 mg protein/mL were prepared and the fluorescence at each concentration measured in a TD-700 fluorometer (Turner Designs, San Jose, CA, USA). Protein concentration was measured using the nanodrop and absorbance at 280 nm. 

### 2.8. Alexa488 Labeling of TetC Constructs

Some TetC constructs were labelled with Alexa Fluor 488 instead of fusion to emGFP. Alexa Fluor™ 488 Protein Labeling Kits from Invitrogen were used to label TetC proteins according to the manufacturer’s protocol. However, since the truncated TetC proteins were smaller than 30 kDa they were purified using Ni-NTA agarose rather than the columns provided. After column washing with 10 mM imidazole in elution buffer, protein was eluted in elution buffer provided in the kit and the addition of 350 mM imidazole. All eluted proteins were concentrated and the buffer exchanged against PBS. Protein concentration was determined by absorbance at 280 nm. Labelled constructs were stored at 4 °C.

### 2.9. TetC Staining and LSCM Imaging

Muscles were dissected in MPS. Dissected preparations were pinned by their tendons in petri dishes lined with Sylgard (Dow, Dewsbury, UK) then incubated with 5–200 µg/mL GFP-865:TetC or other emGFP-TetC constructs for 20–60 min at room temperature, followed by at least one 15 min wash. We confirmed that emGFP alone did not stain NMJs (data not shown). Some muscles were counterstained by incubation for 10–20 min in either TRITC-α -BTX or Alexa647-α-BTX (Invitrogen-Life Technologies, Paisley, UK; 2–5 µg/mL in MPS). The muscles were then washed twice with MPS for 10 min and imaged using either an Olympus BXII epifluorescence microscope via Hamamatsu C5810 or Orca ER cameras interfaced to Openlab (Improvision, Coventry, UK) software running on an Apple Mac PowerPC, or using a BioRad Radiance 2000 laser scanning confocal microscope (LSCM; Biorad, Hemel Hempstead, UK) via a 40× water dipping objective lens mounted on a Nikon Eclipse E600FN epifluorescence microscope. Specimens were scanned with Argon (488 nm), HeNe (543 nm) or red diode lasers (637 nm), normally at 500 lps, and images z-series were captured at either 512 × 512 or 1024 × 1024 pixels resolution. Some images were obtained using a Zeiss LSM 880 confocal microscope and z-series rendered in 3D using Imaris 9.6.1 (Bitplane, Zurich, Switzerland).

For quantification of nerve terminal fluorescence with increasing concentration of GFP-865:TetC, images were captured in the epifluorescence microscope at fixed exposure and camera gain and pixel intensity, averaged in regions of interest enclosing a motor nerve terminal, at 20 min intervals using Openlab software, while progressively increasing the concentration of GFP-865:TetC (15 min washes before imaging at each concentration).

Fluorescence recovery after photobleaching (FRAP) was carried out in the Radiance 2000 confocal microscope using an inbuilt procedure in Lasersharp software. A reference image was first obtained, then the argon laser was scanned at maximum laser power on a small (approximately 50 × 50 pixels) region of interest (ROI) covering part of the terminal. Subsequent 512 × 512 pixel images of the whole terminal were obtained at 500 lps with constant, reduced laser power (1% of maximum), at 10 min intervals over the following hour. Fluorescence intensity of the bleached region was expressed as a percentage of the maximum intensity at the same laser power setting, before bleaching.

### 2.10. Tension Measurements and Electrophysiology

For isometric tension recording, isolated mouse FDB muscles were pinned by their distal tendons to the base of a Sylgard-lined chamber and bathed in MPS (10 mL). The proximal tendon was attached to an ML102 force transducer (AD Instruments, Oxford, UK) and the tibial nerve was aspirated into a capillary glass-tipped suction electrode connected to a DS2 Stimulator (Digitimer, Welwyn Garden City, UK), then stimulated with pulses 0.1–0.2 ms in duration and nominally up to 10 V amplitude at frequencies of 0.5–50 Hz. Stimulus trains (2–50 Hz, for up to 2 s) were triggered via a Powerlab 26T interface (AD Instruments), which was also used to measure force production, using Labchart 7 software (AD Instruments, Oxford, UK) running on an Apple iMac computer. Muscles were either preincubated in emGFP-TetC constructs (20–100 µg/mL) for 30 min then transferred to the chamber and recordings made in MPS, or the constructs were added to MPS in the recording chamber and recordings made 30–60 min later, with or without subsequently washing the preparations. Analysis was carried out offline using Labchart 7/8.

Intracellular recordings were made from mouse FDB or ETA preparations pinned by their distal and proximal tendons to a Sylgard-lined recording chamber and the muscle nerves stimulated using suction electrodes. Muscle action potentials and contractions were blocked by preincubating the muscles in µ-conotoxin GIIIB (µCTX-GIIIB; Sigma-Aldrich, Glasgow, UK) at a concentration of 1–2 µg/mL in MPS for 20–30 min. Microelectrodes were pulled with a Sutter P87 puller, filled with 3 M KCl (resistance typically 20–50 MΩ) and mounted on an MP-85 manipulator (WPI-Europe, Hitchin, UK). FDB muscle fibres are less than 1 mm in length and therefore isopotential, so it was not necessary to position the electrode tip precisely in the endplate region in order to record endplate potentials with high fidelity [13,38]. For ETA recordings, NMJs were initially located using differential interference contrast microscopy and subsequently by fluorescence microscopy after labelling them with full length emGFP-865:TetC. Labelled NMJs were imaged using a Zeiss LSM 880 confocal microscope. In either FDB or ETA muscles, membrane potential and endplate potential (EPP) recordings were obtained using an Axoclamp 2B or 900A amplifiers connected to a CED Micro1401 interface (Cambridge Electronic Design, Cambridge, UK) or Digidata 1550B [Molecular Devices, San Jose, CA, USA) and digitised at 50 kHz using a PC running WinWCP (Strathclyde Electrophysiological Software, Glasgow, UK), Spike2 (Cambridge Electronic Design) or ClampEx (Molecular Devices) software. Stimulation, via a Digitimer DS2 stimulator (up to 10 V, 0.2 ms), was triggered either using the Micro1401 or Digidata interfaces, or a Digitimer 4030 Programmer. Nerve-evoked endplate potentials (EPPs) or spontaneous miniature EPPs (MEPPs) were recorded either from preparations pre-incubated in emGFP-TetC constructs or after adding the emGFP-TetC constructs to the recording chamber (20–100 µg/mL in MPS). Analysis of EPPs and MEPPs was carried out offline using either WinWCP, Minianalysis (Synaptosoft, Atlanta, GA, USA), or pClamp 10 (Molecular Devices, Sunnyvale, CA, USA) software. Amplitudes were corrected to a standard membrane potential of −70 mV, assuming a transmitter reversal potential of −5 mV (V_c_ = V_o_.65/(E_m_-5); where V_c_ is the corrected voltage, V_o_ the observed amplitude and E_m_ the positive (inverted) value of the observed resting membrane potential), before graphical summary and statistical comparison.

### 2.11. Confocal Endomicroscopic (CEM) Imaging

CEM imaging was carried out using either single-waveband (488 nm diode laser excitation) or dual-waveband (488 nm and 660 nm diode laser excitation) Cellvizio^®^ instruments (Mauna Kea Technologies, Paris, France) fitted with 1.5 mm diameter, Proflex S-1500 optical fibre probes. Pilot experiments were performed initially at the Kroto Research Institute, Sheffield using a dual waveband CEM instrument (DW-CEM) in this facility, with the assistance of Dr Nicola Green, and continued in Edinburgh with a different DW-CEM system rented from Mauna Kea Technologies. Probes were calibrated in accordance with the manufacturer’s instructions. For DW-CEM, preparations were simultaneously incubated for 20–30 min in Alexa647-α-bungarotoxin (5–10 µg/mL) and either full length emGFP-865:TetC or truncated constructs (20–50 µg/mL) then washed for 20 min in MPS. The tip of the CEM probe was applied to muscle surfaces, as described previously [20,39]. Image sequences (videos) were captured in real time at 12 frames per second (fps) and individual image frames were extracted and saved in .jpg, .png or .tiff formats. Image sequences were processed in FiJI, downloaded from https://imagej.net/software/fiji/ (accessed on 10 September 2021). In some instances, images of NMJs were stabilised and aligned using the StackReg/TurboReg plugins, downloaded from http://bigwww.epfl.ch/thevenaz/stackreg/ (accessed on 10 September 2021).

### 2.12. Statistics and Graphics

Quantitative data were statistically analysed and graphed using software tools in Microsoft Excel and Prism 7 (Graphpad, San Diego, CA, USA).

## 3. Results

We describe first conventional microscopy of NMJs labelled with fluorescent derivatives of the tetanus toxin C-fragment (TetC). Second, we show from functional and electrophysiological recordings that labelled TetC constructs do not impair neuromuscular transmission, suggesting they may be safe to use in vivo. Third, we describe visualisation of nerve terminals stained with fluorescent TetC constructs using single- or dual-waveband CEM.

### 3.1. GFP-TetC Constructs Selectively Stain Motor Nerve Terminals

We initially tested a clone of the complete tetanus toxin C-fragment, also known as the Hc component of the heavy chain, comprising 458 amino acids numbered from amino acids 857–1315 in the full-length toxin. This construct therefore excluded both the toxic light chain (A fragment) and the membrane translocation region (B or Hn fragment) of the full-length tetanus toxin protein [30]. We tagged and co-expressed the C-fragment with Green Fluorescent Protein at the N-terminus (GFP-TetC). Incubation of isolated mouse muscles with GFP-TetC for 20–40 min, at concentrations ranging from 5–100 µg protein/mL produced selective, concentration-dependent labelling of motor nerve terminal membranes, with high contrast and low levels of background fluorescence (Figure 1). Conventional laser scanning confocal microscopy (LSCM) of unfixed, vitally-stained preparations revealed binding of GFP-TetC both diffusely and at punctate aggregates (‘hotspots’; Figure 1A, inset) in the motor nerve terminal membrane, as indicated also in other studies [28]. This pattern of staining was stable at room temperature for at least 4 h. In preparations from adult mice, although short lengths of the preterminal axon were also stained, myelinated regions of the axons were not labelled (Figure 1A–C, arrows). We attempted post-fixation immunostaining of preparations with neurofilament and SV2 after imaging junctions vitally stained with GFP-TetC. However, the GFP-TetC staining of nerve terminals did not endure fixation very well, so our attempts to co-localise TetC and Sv2 were not successful.

We investigated the stability of bound GFP-865:TetC further by measuring fluorescence recovery after photobleaching (FRAP) of regions of interest (ROIs) enclosing part of the nerve terminal. After bleaching these circumscribed ROIs, partial recovery of fluorescence occurred in the bleached region with a recovery time constant of 14.3–15.5 min, suggesting that the binding sites for GFP-865:TetC were mobile, but unlikely to be freely diffusible within the plane of nerve terminal plasma membranes (Figure 2). Recovery of fluorescence also occurred in bleached regions containing hot-spots (recovery 50–80%, time constant 22–44 min, n = 3 NMJs/muscles), suggesting that tetanus-toxin binding sites were also readily recruited into these sites. We did not investigate the mobility of TetC binding sites further in the present study.

### 3.2. Truncated emGFP-TetC Constructs Also Stain Nerve Terminals

Next, we modified GFP-TetC, first by substituting GFP with a brighter fluorescent protein variant, “Emerald” fluorescent protein (emGFP) [31]. The constructs we made and examined for fluorescent staining of motor nerve terminals are illustrated in Figure 3A–C. We refer to our constructs with reference to the N-terminal amino acid to the amino acid location in full length tetanus toxin. Thus, from hereon we refer to the full length (FL) C-fragment as 865:TetC, and the N-terminally truncated fragments we studied as 1066:TetC, 1093:TetC, 1109:TetC and so on. When we compared the fluorescence of these constructs ligated with either eGFP or emGFP, it appeared that the fluorescence of constructs labelled with emGFP were about twice as bright as those labelled with eGFP (Figure 3D).

Next, we tested the effect of truncating TetC on selective labelling of motor nerve terminals, expressing emGFP at the N-terminus of the truncated construct (Figure 4). The appearance of motor nerve terminals labelled with emGFP was similar to those labelled with GFP or to TetC fragments labelled by conjugation with Alexa 488 (Figure 4A,B). We found that emGFP-1066:TetC, a protein fragment comprising amino acids 1066–1315 only (that is, the 249 C-terminus amino acids of TetC) also produced strong and reliable staining of mouse, rat and human NMJs (Figure 4C). Similar results were obtained with a shorter construct, emGFP-1093:TetC, containing 222 C-terminal amino acids. We did not compare quantitatively the fluorescence intensity of these bound TetC constructs but, consistent with previous reports of improved receptor binding [30], our subjective impression was that fluorescence was brighter with emGFP-1093:TetC and emGFP-1066:TetC compared with full length emGFP-865:TetC. However, we did not test in the present study whether simultaneous staining of different preparations with emGFP-TetC variants produced quantitatively different fluorescent staining intensities. Thus, our subjective impression is subject to quantitative validation. A construct truncated at amino acid 1109 (emGFP-1109:TetC; 206 C-terminus amino acids) also produced strong initial staining of NMJs (Figure 4C) but this was relatively transient; nerve terminal fluorescence intensity and contrast were typically degraded over the following 2–4 h (see below). This was possibly due to increased uptake and axonal transportation but we did not explore the mechanism further in the present study. Constructs containing 178 or fewer C-terminus amino acids (Figure 3B, blue bars) showed no overt NMJ staining at concentrations up to 200 µg/mL (data not shown).

### 3.3. emGFP-TetC Constructs Stain Neonatal Motor Nerve Terminals

TetC fragments also stained motor nerve terminals in neonatal muscle preparations. Figure 5A–D shows images of NMJs in isolated *triangularis sterni* (TS) muscles stained with emGFP-1093:TetC in muscles dissected from neonatal rats aged 3–10 days postnatal (P3-P10). In addition to staining NMJs, emGFP-1093:TetC also revealed the characteristic overgrowth of motor axons normally associated with immature, polyneuronally innervated NMJs [40,41]. By P10, however, axonal staining appeared fainter (Figure 4D) and, as reported above, was largely absent in adult preparation (Figure 5E). It remains unclear whether this decrease then loss of axonal staining is due to postnatal loss of receptor for the TetC ligand in axonal membranes, or to exclusion of the dye from axonal binding sites by the compacted myelin sheath, which is the last to form during NMJ development [42].

### 3.4. emGFP-TetC Constructs Also Stained Other Cell Types

In some preparations, emGFP-Tet constructs produced notable, additional labelling of sparsely distributed axons. However, these axons were not α-motor axons terminating at NMJs. Close inspection of LSCM z-series showed that many of these axons were associated with blood vessels or passed through connective tissue overlying NMJs (Figure 6A–C; Appendix A). We presumed these to be either unmyelinated sensory axons or autonomic motor axons [43,44,45]. There was also no evidence that emGFP-TetC constructs stained either myelinating Schwann cells, terminal Schwann cells, or kranocytes localised to NMJs [13,46]. However, in some preparations unidentified cells resembling fibroblasts or macrophages, and that were not selectively localised to NMJs, were also labelled (Figure 6C,D). We did not investigate these ectopic binding sites further in the present study.

In sum, we found that protein fragments about half of the full length TetC C-terminal fragment retain capacity for selective binding to nerve terminal membranes [30]. When conjugated with GFP or related proteins, these constructs give rise to high-contrast, selective staining of motor nerve terminals. emGFP-TetC constructs also produced additional staining of unmyelinated axons and other, as yet unidentified, cell types.

### 3.5. emGFP-TetC Constructs Do Not Impair Neuromuscular Transmission or Function

Next, we tested whether emGFP-TetC constructs would impair neuromuscular synaptic transmission or function, using isometric force of muscle contractions in response to nerve stimulation and intracellular recordings from individual NMJs.

Figure 7A–D shows twitch and tetanic force responses in wild-type mouse nerve-muscle preparations evoked before and one hour after incubation in emGFP-1093:TetC at 50 µg/mL, a concentration that was sufficient to label motor nerve terminals. There was no discernible effect on nerve-evoked twitch tension or sustained, tetanic muscle force. Consistent with the functional recordings, intracellular recordings from individual NMJs, after a similar period of incubation in either full length or truncated emGFP-TetC constructs, showed no discernible impairment in neuromuscular transmission. For instance, when we measured EPPs and MEPPs from specific NMJs in mouse ETA muscle, before and after incubating preparations with a range of concentrations of emGFP-865:TetC (20–100 µg/mL; Figure 8A,B), there was no significant effect (Figure 8A–E). Similarly, there was no discernible or significant effect of emGFP-1093:TetC on either EPPs or MEPPs recorded from muscle fibres sampled with microelectrodes before, during or after washing emGFP-1093:TetC from media bathing isolated mouse FDB muscles (Figure 8F–I). In these preparations, mean EPP amplitude (corrected to a resting potential of −70 mV) in MPS was 15.5 ± 5.0 mV (mean ± SEM, n = 3 muscles, 18 muscle fibres; 95% CI = 12.4–18.6 mV) and 18.5 ± 5.0 mV after 1 h incubation in 50 µg/mL emGFP-1093:TetC (n = 3 muscles, 16 muscle fibres; 95% CI = 11.8–17.2 mV). One hour after washing preparations with MPS (when terminals remained labelled) mean EPP amplitude was 15.0 ± 7.0 mV (n = 3 muscles, 17 muscle fibres; 95% CI = 9.5–16.2 mV). None of these differences were significant (*p* = 0.74, ANOVA). We did not perform a rigorous quantal analysis in the present study, which requires analysis of synaptic currents under voltage clamp, but it is implicit from these voltage data that evoked vesicular release (quantal content) was also unaffected by incubation in emGFP-TetC constructs. MEPP frequency was also not significantly altered. In MPS, mean MEPP frequency was 1.29 ± 0.38 s^−1^ (Mean ± SEM; N = 4 muscles, n = 68 muscle fibres). After incubation in emGFP-865:TetC, mean MEPP frequency was 2.45 ± 1.62 s^−1^ (N = 4, n = 64; *p* = 0.234, paired *t*-test).

Thus, taking MEPP and EPP measurements together, transmitter release and sensitivity of AChR to neurotransmitter were affected little, if at all, by incubation and labelling in emGFP-TetC conjugates at concentrations up to at least 100 µg/mL. By contrast, labelling postsynaptic acetylcholine receptors (AChR) with fluorescent conjugates of α-bungarotoxin (5–10 µg/mL), reduced then blocked both nerve-evoked muscle contractions and EPPs after incubating muscles for more than 10 min (data not shown).

In sum, our electrophysiological analysis suggests that emGFP-1093:TetC and other TetC constructs, applied at concentrations that produced bright staining of motor nerve terminals, were physiologically inert from the standpoint of functional neuromuscular transmission, underscoring the potential utility of emGFP-TetC constructs as inert vital stains for motor nerve endings, at least in explanted preparations.

### 3.6. CEM Visualises Adult Nerve Terminals Stained with emGFP-TetC Constructs

Next we tested whether motor nerve terminals labelled with emGFP-TetC or Alexa488-TetC constructs would be discernible using 1.5 mm diameter, optical-fibre probes coupled to a single-waveband (488 nm excitation) CEM system. Compared with our previous attempts using other fluorochromes [21], subcutaneous injection of 50–100 µL of labelled emGFP-TetC constructs (20–50 µg/mL) into the hind limbs of anaesthetised mice produced much brighter in situ labelling of NMJs in the vicinity of the injection site that was clearly visible and with high contrast, using CEM (Appendix A). As in our previous studies [20,21], we captured real time image sequences while slowly gliding the optical fibre probe by hand over the muscle surface. With practice, quite stable image sequences (videos) were obtained by this method. However, further compensation for image vibration or drift (‘camera shake’) proved possible by applying the StackReg/TurboReg image alignment (registration) utility for Fiji to the video recordings (see Methods; Appendix A).

For further CEM visualisation of nerve terminal staining, we isolated nerve muscle preparations, stained them by incubation in 20–100 µg/mL emGFP-TetC constructs and imaged surface NMJs. Figure 9A–D shows still frames from CEM videos of the motor nerve terminals in mouse TS preparations stained with full length (emGFP-865:TetC), emGFP-1066:TetC, emGFP-1093:TetC, or emGFP-1109:TetC. Nerve terminal fluorescence was stable for at least 4hr using emGFP-1066:TetC or emGFP-1093:TetC (Figure 9C; see also Appendix A) but marked degradation of staining was evident in images obtained after staining with emGFP-1109:TetC over the same period (Figure 9D). We also successfully used CEM to visualise NMJs labelled with Alexa488 conjugates of full length TetC (Appendix A).

We also examined mouse NMJs 21–28 days after unilateral sciatic nerve crush, when axons and motor nerve terminals were expected to have regenerated. It proved more difficult to locate distal hind limb NMJs with certainty in situ compared with those in unoperated limbs. In isolated preparations, however, nerve terminals were clearly labelled and evidence of nerve-terminal and axonal sprouting was visible (Figure 9E). Subsequent LSCM, after counterstaining endplates with TRITC-α-bungarotoxin, confirmed the reinnervation of most NMJs, with nerve terminal sprouting evident in some of them (Figure 9F–H).

### 3.7. Dual-Waveband CEM Resolves NMJ Pathology in Real Time

We asked next whether dual-waveband CEM (DW-CEM) would enable detection of abnormal motor innervation of muscle. We answered this by simultaneously visualising emGFP-TetC constructs bound to presynaptic nerve terminal membranes and a fluorescent conjugate of α-bungarotoxin bound to postsynaptic motor endplate membranes.

As proof of concept, we first applied DW-CEM to isolated preparations from *thy1.2YFP16* transgenic mice, in which motor axons and nerve terminals are endogenously fluorescent due to expression of YFP in motor neurons. We had shown previously that single waveband CEM readily detects normal and abnormal motor nerve terminals in these mice [20,21]. Since the long-wavelength laser excitation in the Cellvizio^®^ DW-CEM is at 660nm, we counterstained AChR in these preparations with the long wavelength emission conjugate, Alexa647-α-BTX (fluorescence pseudocoloured red or magenta in Figure 10 and Appendix A). All NMJs showed short- and long- wavelength fluorescence, respectively, co-localised to NMJs.

We then used DW-CEM to examine isolated mouse muscles stained with emGFP-TetC constructs (Figure 10D–F; Appendix A). For these tests, we incubated hind limb muscles (typically EDL, SOL or FDB) or TS muscles in either full length emGFP-865:TetC or the strongly-binding variants emGFP-1066:TetC or emGFP-1093:TetC (20–100 µg/mL). Simultaneously, we stained postsynaptic AChR with Alexa647-α-BTX (5–10 µg/mL). As in preparations from mice with YFP expression in axon terminals, all endplates showed co-localisation of nerve terminal and endplate fluorescence. Similar results were obtained applying DW-CEM to dual-stained rat NMJs (data not shown). We also attempted to locate NMJs in human muscle tissue explants with DW-CEM, since nerve terminals stained with emGFP-TetC constructs in these explants were visible using LSCM (see Figure 4). However, the combination of relatively small size of human NMJs [35,47,48] and relatively low resolution of DW-CEM, together with high non-specific background fluorescence due to staining of connective tissue, evidently precluded reliable identification of NMJs in the human muscle samples supplied, using this method. This was the case with NMJs stained either with full-length emGFP-865:TetC or with the truncated constructs emGFP-1066:TetC or emGFP-1093:TetC.

We reverted to mice in order to assess whether pathological abnormalities of motor innervation were distinguishable with DW-CEM. We imaged NMJs in two lines of mice in which nerve terminals undergo slow, asynchronous degeneration (Figure 11), specifically, NMJs in young adult *Wld^S^* mutant mice following axonal injury and spontaneous degeneration of terminals in SOD1G93A mice, a model of ALS [12,20]. In *Wld^S^* mice, we visualised NMJs in isolated hind limb muscles 5–6 days after unilateral sciatic nerve section. Our previous studies established that only about half the axotomised NMJs undergo degeneration or withdrawal of motor nerve terminals from motor endplates during the first 7 days after axotomy. The Ube4b-Nmnat2 chimeric protein expressed in *Wld^S^* motor neurons continues to protect the remaining motor nerve terminals, resulting in complete or partial occupancy of their motor endplates [12,39,49,50].

DW-CEM imaging of *Wld^S^* mouse muscles performed 5–6 days after axotomy clearly distinguished denervated from innervated NMJs (Figure 11A,C,E; Appendix A). Denervated NMJs showed only Alexa647-α-BTX fluorescence, while the remaining, innervated NMJs showed co-localised emGFP-1093:TetC fluorescence. However, the limited spatial resolution of DW-CEM made it difficult to be certain of differences in the fractional occupancy of NMJs; that is, NMJs from which motor nerve terminals may have been undergoing progressive degeneration or withdrawal [12]. Contralateral muscles stained with emGFP-1093:TetC and Alexa647-α-BTX showed 100% innervation, similar to that shown in Figure 10D–F (data not shown).

A similar mosaic of denervated and innervated NMJs was apparent using DW-CEM to examine muscles dissected from liminally symptomatic SOD1G93A mice. Previous studies have shown that NMJs undergoing spontaneous, progressive denervation in these mice, associated with loss of muscle function [10,11,20,51,52,53,54]. As in axotomized *Wld^S^* mouse muscles, denervated and innervated endplates were readily distinguished by DW-CEM (Figure 11B,D,F; Appendix A). Muscles dissected from SOD1G93A littermate controls showed 100% innervation of NMJs (data not shown).

Thus, vital staining of nerve terminals with TetC and endplates with α-bungarotoxin combined with DW-CEM enabled resolution of denervated from innervated NMJs in two animal models of motor neuron disease. However, further refinements to the spatial resolution of the CEM method are ultimately required in order to translate this methodology to visualisation of normal or diseased NMJs in humans.

## 4. Discussion

Our principal goal in this study was to find effective, physiologically-inert labelled peptides that could be used for vital fluorescent staining of motor nerve terminals at NMJs, with sufficient contrast for visualisation using minimally invasive live imaging techniques such as fibre-optic CEM [20,21]. We utilised the tetanus toxin C-fragment for this purpose, noting also a previous report that truncations of the C-fragment appear to bind more effectively to their ganglioside receptors with higher affinity than the full-length fragment [30]. The Results show that incubation of NMJs in rodent muscles with fluorescent constructs greater than about half the length of the C-fragment of tetanus toxin (that is, up to 206 amino acids from the C-terminus) are readily and stably stained and visualised, using either conventional fluorescence microscopy or CEM. Acutely, murine motor nerve terminals labelled with the TetC fragments showed no overt deficiency in neuromuscular transmission, indicating possibilities for potentially important applications, combining morphological and physiological investigation of NMJ structure and function [55,56,57,58]. We also extended our previous investigations of the utility of CEM [20,21] by demonstrating that DW-CEM enables live imaging of motor innervation by simultaneous visualisation of presynaptic terminals (labelled with emGFP-865:TetC) and postsynaptic receptors (labelled with Alexa647-BTX). DW-CEM also readily resolved intact from degenerating motor nerve terminals in mouse models of progressive axonal and neuromuscular synaptic degeneration.

### 4.1. Binding of GFP-TetC Derivatives

GFP-TetC derivatives (or Alexa488-TetC conjugates) appear superior in several respects to previous compounds utilised as non-toxic vital stains for NMJs, including nerve terminal staining using GFP-tagged constructs of the botulinum toxin heavy chain (BoNT-HC), which we presume to be due to a greater density of binding sites for TetC than for BoNT-HC [21]. There are also benefits compared with simpler organic molecules such as styryl dyes like 4-di-2-Asp or FM1-43. For instance, 4-di-2-Asp staining of NMJs is very effective when viewed with conventional fluorescence microscopes but background staining is relatively high. Likewise, staining with FM1-43 typically produces significant background staining, exacerbated by staining of intracellular membranes in any damaged cells [58,59,60,61]. FM1-43 staining also requires nerve stimulation or depolarisation during loading in order to generate contrast with passive background staining [57,59,62]. However, FM1-43 and related dyes have the additional attribute of reporting nerve function, since nerve stimulation of FM1-43 labelled nerve terminals ‘destains’ them at rates that reflect synaptic vesicle recycling [57,63], which our TetC derivatives do not.

Mutational analysis of the C-fragment suggests that sites for ganglioside sialic acid binding motifs are located between amino acids 1271–1282, 1214–1219 and a key residue, amino acid 1290, of the TetC fragment [36,64,65,66]. It is therefore perhaps surprising that we saw no staining with constructs clipped at amino acids 1137 or 1214. The same sites on TetC were reported recently to bind to nidogens, protein constituents in motor nerve terminals [34]. Perhaps interference between GFP or emGFP, when tagged to the N-terminus of our TetC constructs in close proximity to these sites interfered with binding to target molecules on presynaptic terminals. Consistent with this hypothesis, we found that emGFP-1109:TetC binding appeared less resilient than the longer constructs emGFP-1066:TetC or emGFP-1093:TetC. By contrast, these two GFP-based fluorochromes appeared, subjectively, to give even brighter staining than full length emGFP-865:TetC, consistent with reports that similar truncations of TetC increase binding to target gangliosides [30]. The complexes formed between TetC constructs and their binding sites in nerve terminal membranes were evidently diffusible or otherwise mobile in the presynaptic membrane, since our measurements of FRAP (Figure 2) indicated movement of GFP-TetC receptor complexes into the bleached region. Based on Equation (1) given by Gaffield et al. [67], the mobility for the TetC-receptor binding complexes appeared to be orders of magnitude less than that expected for passive diffusion but more detailed measurements are required to establish an accurate diffusion coefficient. A critical 47 amino acid sequence required for binding of the truncated fragment appears to lie between the amino acids at positions 1093 and 1140. However, this region alone was not sufficient for binding. Constructs lacking 10 amino acids from the C-terminus do not bind to ganglioside receptors in neuronal membranes [30].

An unexplained feature of staining with all our emGFP-TetC derivatives was that in most preparations there was little or no axonal staining, or axonal staining was limited to short lengths of preterminal axons, presumably those extending beyond the last Schwann cell heminode [13,14]. However, in neonatal preparations there was overt axonal staining (see, for example, Figure 5), and unmyelinated sensory or autonomic axons were overtly stained in other, adult preparations (see Figure 6). It remains unclear whether the absence of complete motor axon staining in adult preparations is due to absence of tetanus toxin receptors or failure of our emGFP-TetC constructs to penetrate the extracellular spaces restricted by the presence of myelin sheaths. However, the apparent absence of nodal staining suggests that absence of protein or ganglioside receptors in these regions is a more likely explanation.

Finally, although our physiological analysis indicated no discernible acute effects of emGFP-1093:TetC conjugate on neuromuscular physiology, further investigation is required to establish any potential longer term consequences of administration of fluorescent TetC derivatives for the integrity of motor neurons, or other cell types.

### 4.2. Limitations of CEM and Future Prospects

In the context of NMJ imaging, the main strengths of CEM lie with the flexibility and small diameter of the optical fibre probes, less than that of some muscle biopsy needles, together with the ability to capture image sequences at relatively rapid, smooth rate (12 fps). These features enable rapid appraisal of NMJ disposition and integrity. Thus, the principal advantages of fibre-optic CEM are capability for in situ, real time imaging of fluorescent structures, using a flexible, hand-held miniature probe. Potentially, CEM enables minimally invasive live imaging of NMJs in any muscle via a small (<0.5 cm) skin incision, as we have shown in our previous studies [20,21]. This is not possible using conventional LSCM. Potential applications include rapid appraisal of neuromuscular synaptic pathology and electrophysiological characterisation of synaptic structure-function relationships [8,35,68].

However, the short (near zero) working distance and relatively low spatial resolution of CEM, (constrained by the spacing between optical filaments that make up the optical probes), are shortcomings that limit image quality and the capacity to resolve small NMJs or detailed morphology including individual boutons or fine terminal or axonal sprouts. The optical path length also limits sensitivity and, therefore, bright fluorochromes provide the best opportunities for distinguishing normal or pathological structure. Moreover, we were unsuccessful in visualising NMJs in human explants using CEM, even using the truncations emGFP-1066:TetC or emGFP-1093:TetC. Further advances in minimally invasive or non-invasive live imaging may therefore be required to fully capitalise on the NMJ staining capability of fluorescent TetC constructs, or other molecules with high affinity, to visualise human NMJs in situ. However, emGFP-TetC staining may enable location of NMJs in motor point biopsies using conventional wide-field fluorescence microscopy or LSCM, facilitating their investigation with electrophysiological techniques [2,9,47,48]. The present findings also endorse the possibility that truncated TetC proteins might be effective vehicles for focal delivery of potential therapeutic agents, for uptake into motor nerve terminals and transport to motor neuron cell bodies as novel treatments for ALS or other neuromuscular diseases [26,69,70,71]. Finally, we could distinguish innervated from denervated NMJs using DW-CEM in rodent muscle, so the technique is a potentially valuable one for evaluating the extent of early signs of neuromuscular disease, at least in animal models of disease, for example, the SOD1G93A mouse model of ALS, and perhaps other models as well [16,72].

## 5. Conclusions

Fluorescent derivatives of TetC are useful vital stains for NMJs in species ranging from rodents to humans. If used in conjunction with low concentrations of fluorescent conjugates of BTX, even relatively high concentrations of TetC stain NMJs without inhibiting neuromuscular transmission. However, significant challenges remain before the feasibility of using this technique in a clinical context, in human subjects or patients, can be evaluated in situ. Nevertheless, the fluorescence intensity of TetC conjugates are sufficient for visualisation of NMJs using confocal endomicroscopy (CEM), at least in animal models of disease. For instance, dual waveband CEM resolves innervated from denervated NMJs, counterstained with fluorescent α-bungarotoxin, in mouse models of nerve degeneration or neuromuscular disease. 

## Figures and Tables

**Figure 1 biomolecules-11-01499-f001:**
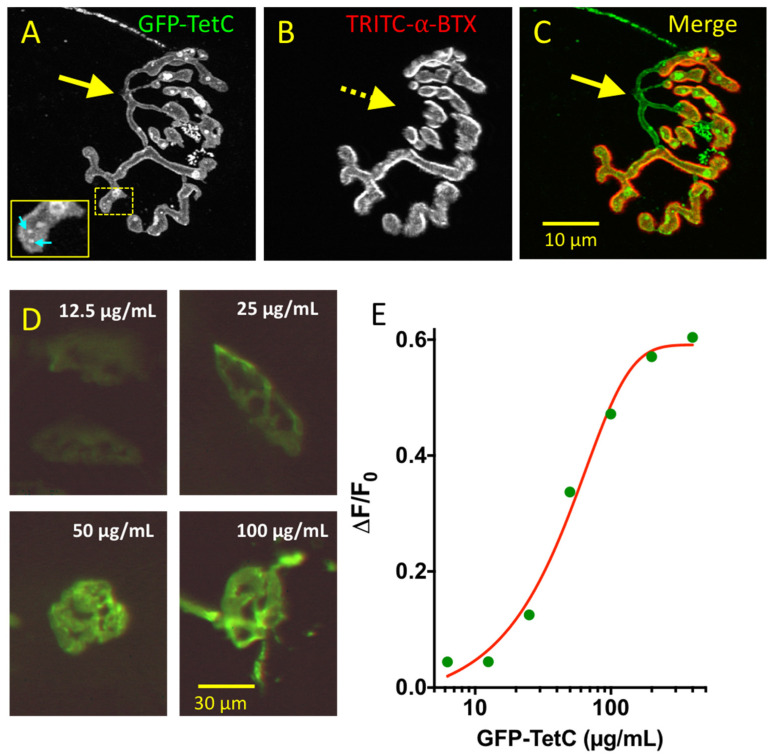
GFP-tagged TetC staining of motor nerve terminals is concentration dependent. (**A**–**C**): LSCM images of an NMJ in an unfixed mouse TS muscle vitally stained with 100 µg/mL of full-length GFP-TetC and 5 µg/mL TRITC-α-bungarotoxin. (**A**): GFP-TetC uniformly stained motor nerve terminals but not their preterminal axons. Both diffuse and concentrated staining (‘hotspots’; arrows and inset) was typically observed; (**B**): Counterstaining with TRITC-α-bungarotoxin staining of postsynaptic AChR; (**C**): Merged image showing alignment of presynaptic staining with slight extension of AChR staining beyond the margins of the nerve terminal staining. (**D**): Wide-field fluorescence images of motor nerve terminals in mouse FDB muscles stained with progressively increasing concentrations (12.5–100 µg/mL, as indicated) of GFP-TetC. (**E**): Graph of ratio of nerve terminal fluorescence to background fluorescence (ΔF/F_0_) versus GFP-TetC concentration. Fluorescence ratio was calculated from the difference between average intensity in a region of interest (ROI) including a continuous portion of the NMJ (range of areas 575 to 1650 pixels) selected using the ‘magic wand’ tool in Fiji, compared with that in a background region outside the NMJ. Discernible staining of NMJs was observed with concentrations about 10 µg/mL but brighter, higher contrast staining was observed at concentrations between 50–100 µg/mL.

**Figure 2 biomolecules-11-01499-f002:**
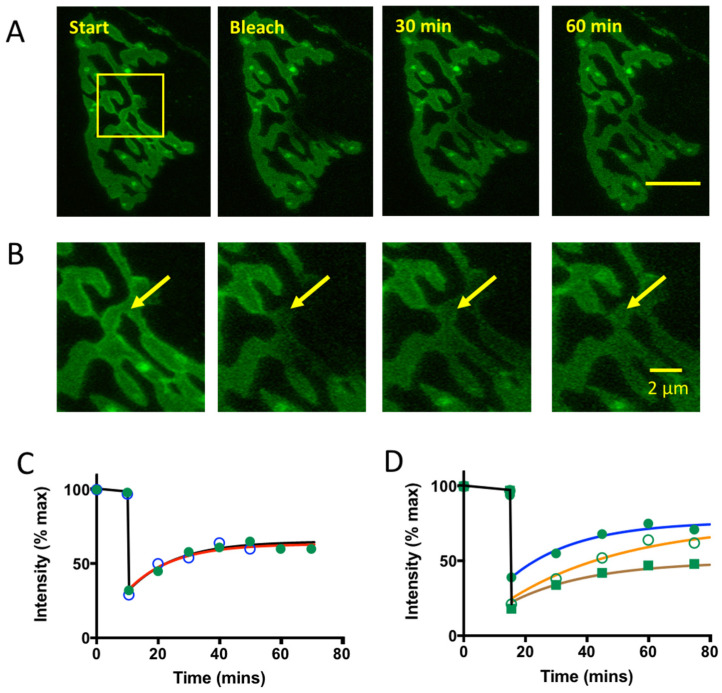
TetC binding sites are mobile within the nerve terminal membrane. (**A**): Sequence of LSCM images of an unfixed mouse TS NMJ showing recovery of fluorescence after photobleaching (FRAP) of the region indicated (yellow outlined box) in the first image of the sequence. Subsequent images were captured at fixed laser power at the times indicated during the following 60 min. (**B**): Images of the showing recovery of fluorescence in the expanded region indicated in A. Yellow arrow indicates the bleached region in this NMJ. (**C**): Graphs of FRAP from two separate experiments (preparations from different mice) based on measurements in diffusely stained regions of the motor nerve terminals. (**D**): Graphs of FRAP from three different experiments (preparations from different mice) based on measurements in punctate-stained, hotspot regions of motor nerve terminals. The relatively slow rates of FRAP suggest a slower process of redistribution of labelled TetC receptor proteins than expected by passive diffusion.

**Figure 3 biomolecules-11-01499-f003:**
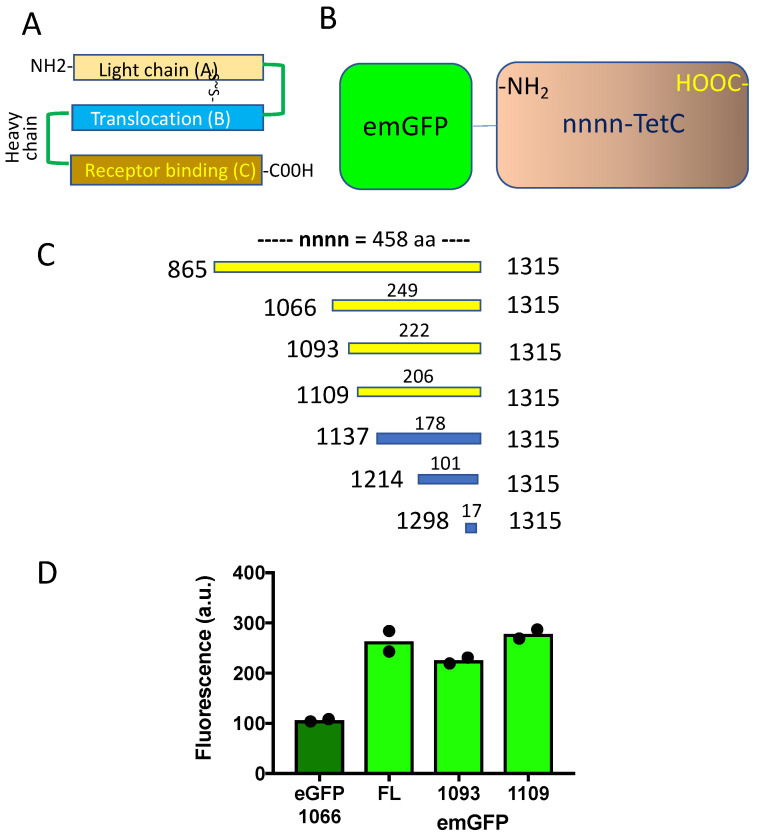
Truncated TetC constructs labelled with emGFP are brightly fluorescent. (**A**): Schematic diagram showing the three principal domains of intact tetanus toxin, including the toxic light chain (A) and inert heavy chain, comprising Translocation (B) and Receptor binding (C) domains. (**B**): Schematic showing association of emerald Fluorescent Protein (emGFP) with TetC constructs made in the present study, comprising variable numbers of amino acids from their truncated N-terminuses (nnnn-TetC); areas drawn in proportion to the molecular weights of GFP and TetC. (**C**): Full length TetC contains 458 amino acids, numbered 865–1315 at the C-terminus of complete tetanus toxin (light + heavy chains). In the present study, we compared truncations at amino acid 1066–1298, as indicated. Yellow bars indicated positive staining of motor nerve terminals (see Figure 4), blue bars indicated no staining. (**D**): Fluorometric data (duplicated) comparing fluorescence of truncated TetC constructs labelled either with enhanced GFP (eGFP-1066) or with emGFP (emGFP-FL = full length; emGFP-1093; or emGFP-1109, referring to constructs emGFP-865:TetC; emGFP-1066:TetC, emGFP-1093:TetC or emGFP-1109:TetC). emGFP labelled constructs were about twice as bright at the same protein concentration as eGFP labelled constructs. Filled circles are resulrs of individual fluorometric measurements, bars are the mean of these in each instance.

**Figure 4 biomolecules-11-01499-f004:**
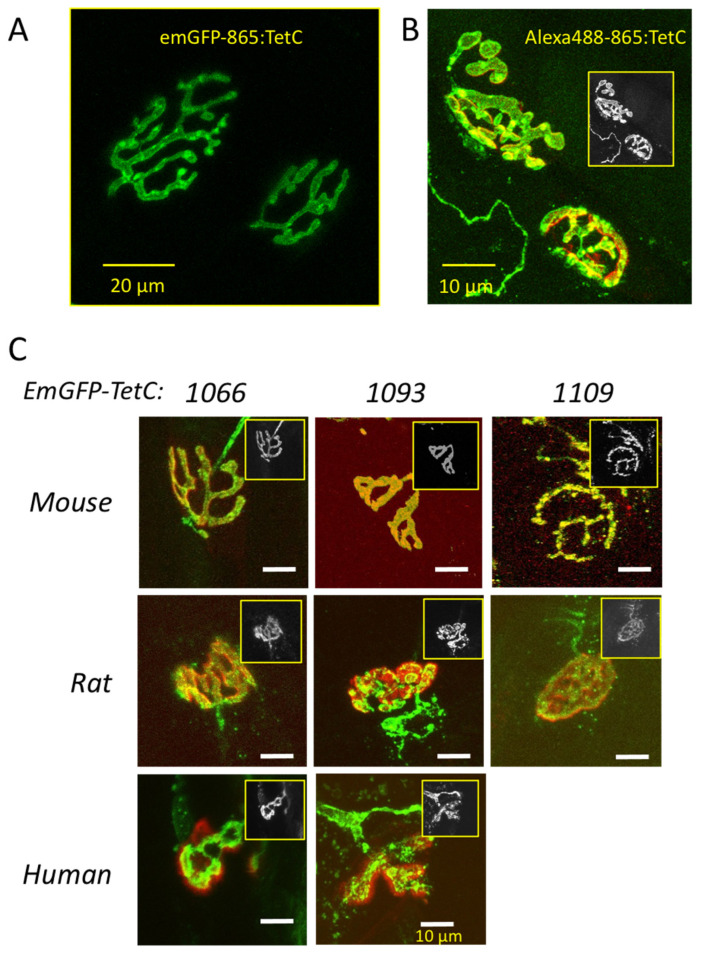
emGFP-TetC derivatives stain rodent and human motor nerve terminals. (**A**): LSCM image of unfixed motor nerve terminals in mouse TS muscle vitally stained with 50 µg/mL emGFP-865:TetC (full length TetC); (**B**): merged LSCM mouse TS NMJs stained with an Alexa488-tagged full length TetC construct (green fluorescence) and AChR counterstained with TRITC-α-bungarotoxin (red fluorescence; colocalised staining yellow); (**C**): Merged LSCM images of NMJs counterstained with TRITC-α-bungarotoxin and TetC constructs truncated at the N-terminal amino acids indicated, in mouse, rat and human muscle explants. All the constructs indicated produced NMJ staining (staining of human explants with emGFP-1109:TetC was not attempted). Insets in (**B**,**C**) show the fluorescent green channel (displayed as grey) for each image.

**Figure 5 biomolecules-11-01499-f005:**
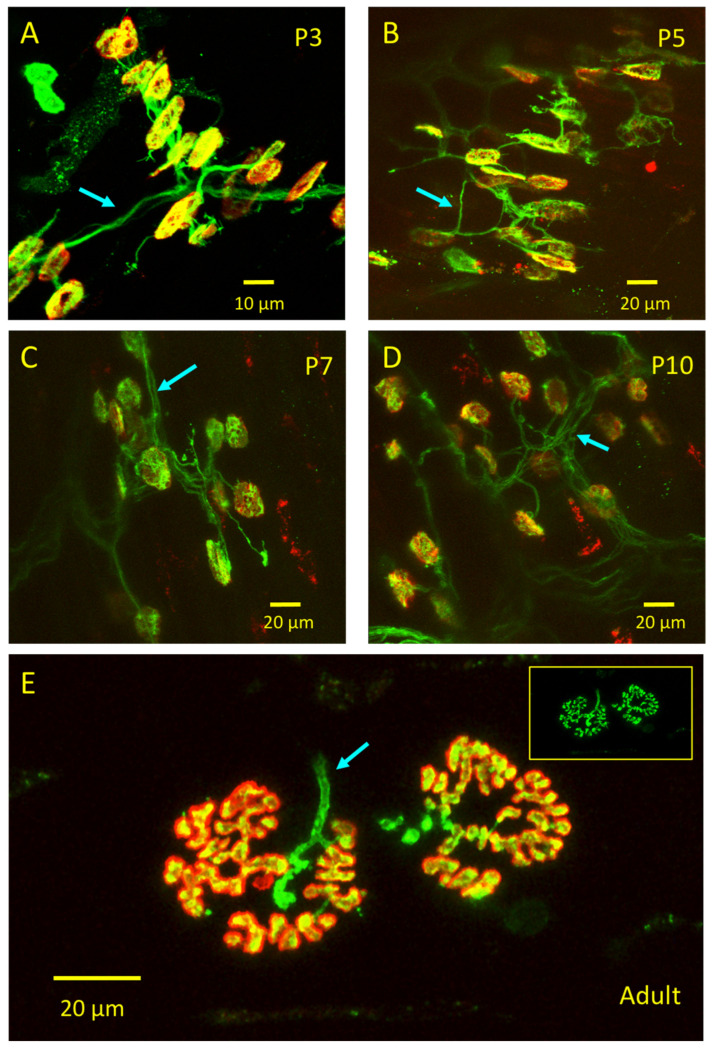
emGFP-TetC constructs also stain neonatal NMJs. (**A**–**D**): LSCM images of the motor innervation of unfixed TS muscle preparations from rats of the postnatal ages (in days, P3–P10) indicated, co-stained with emGFP-1093:TetC and TRITC-α-bungarotoxin. Motor nerve terminals (yellow) and axons (green) that were polyneuronally innervating NMJs were visible at this age. Arrows indicate staining of intramuscular and preterminal axons. (**E**): By contrast, emGFP-1093:TetC intensely stained motor nerve terminals (inset) in an adult (about 150 g) rat TS preparation but, as in adult mice, the preterminal axon (cyan arrow) appeared largely unstained.

**Figure 6 biomolecules-11-01499-f006:**
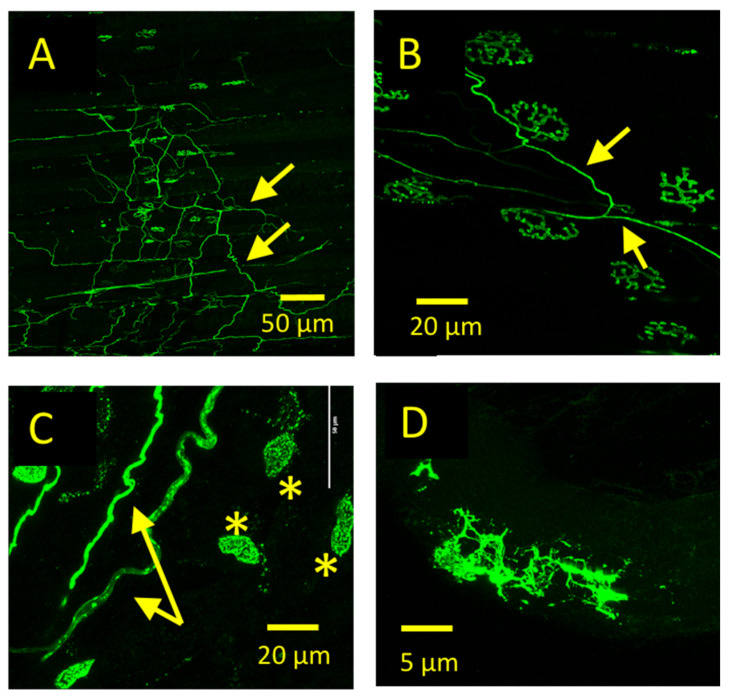
emGFP-TetC constructs also stain unmyelinated axons and unidentified non-neuronal cells. (**A**): Low power LSCM image of an unfixed mouse ETA preparation vitally stained with emGFP-865:TetC. In addition to nerve terminal staining, a network of axonal staining (arrows) was also visible, not terminating at NMJs and likely to be unmyelinated axons of intramuscular sensory axons or vascular autonomic motor axons (see also Appendix A). (**B**): Higher power image of the same preparation as A, showing probable unmyelinated autonomic/sensory axonal staining. Note that although these axons in this maximum intensity projection LSCM image may appear to end on NMJs, inspection of the image z-series indicated clearly that they were not (images not shown); (**C**,**D**): In addition to axonal staining (arrows), in some preparations emGFP-TetC constructs evidently stained unidentified non-neuronal cells (asterisks), possibly macrophages.

**Figure 7 biomolecules-11-01499-f007:**
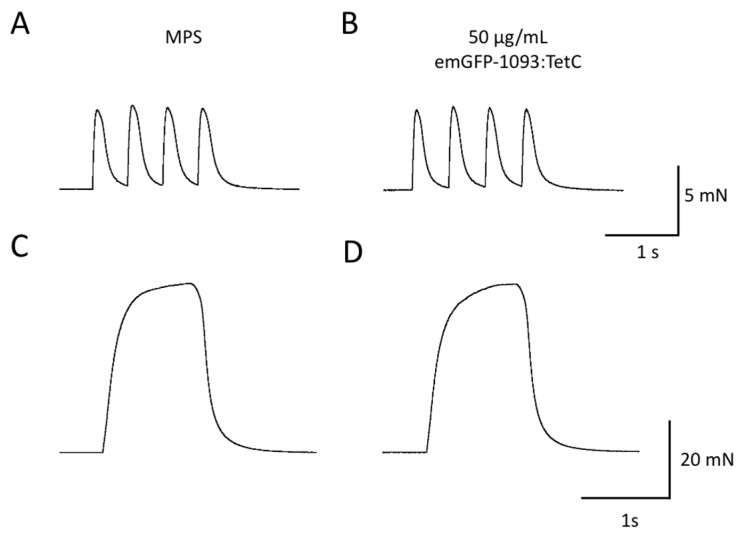
emGFP-TetC constructs have no discernible effect on neuromuscular synaptic function. (**A**–**D**): Isometric muscle force recordings of FDB muscle responses to either low frequency (2 Hz) stimulation producing a train of four twitch responses (**A**,**B**) or high frequency (50 Hz, 2 s) stimulation producing fused tetanic contractions (**C**,**D**). There were no discernible differences in the response characteristics before (in mammalian physiological saline, MPS) or after incubating for 30 min in emGFP-1093:TetC (50 µg/mL). Similar responses were seen with incubation in other emGFP-TetC constructs (not shown).

**Figure 8 biomolecules-11-01499-f008:**
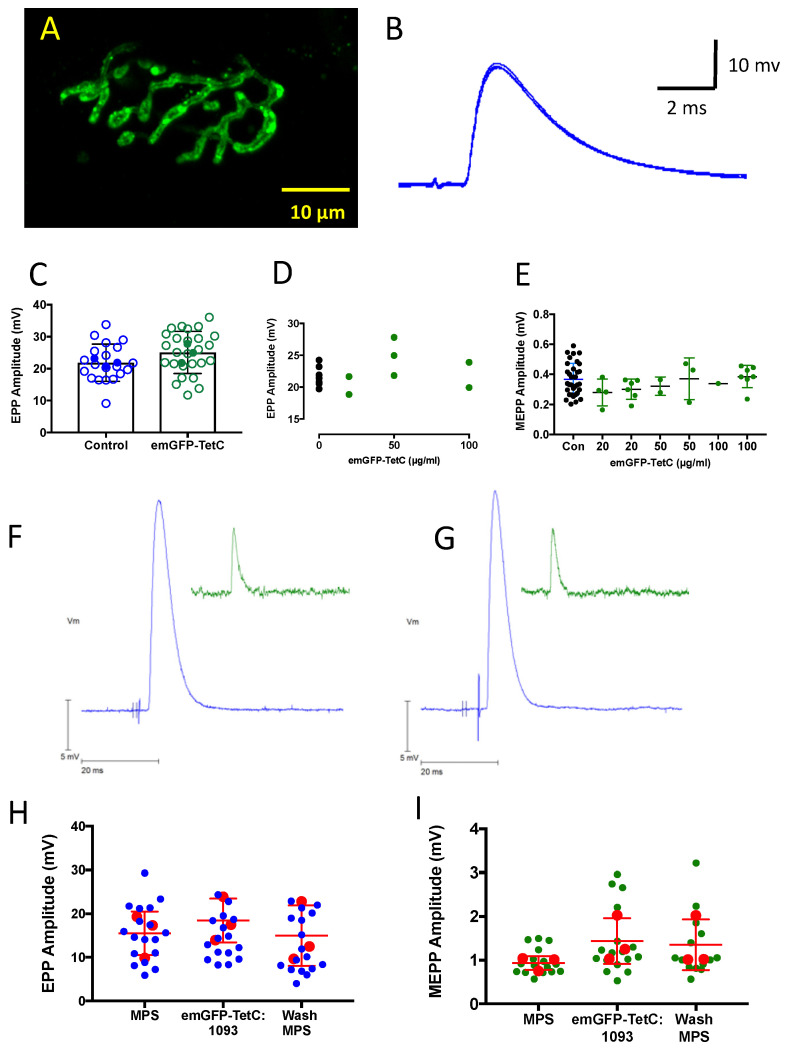
emGFP-TetC constructs do not impair neuromuscular synaptic transmission. (**A**)**:** LCSM confocal image of an unfixed, vitally stained motor nerve terminal in a mouse ETA muscle after incubation for about 30 min in emGFP-865:TetC (50 µg/mL). (**B**): Intracellular recording of an EPP from the same NMJ as shown in A, about 30 min after washing emGFP-865:TetC from the recording chamber. (**C**–**E**): Analysis of EPP (**C**,**D**) or spontaneous MEPP (**E**) recordings from specific labelled NMJs in isolated ETA muscles. (**C**): open circles represent amplitudes of EPPs recorded in individual muscle fibres from three preparations, means from each muscle shown by filled circles. (**D**): Each point represents the mean EPP responses from one muscle (individual mice) following incubation in concentrations ranging from 20–100 µg/mL emGFP-865:TetC (see Methods). (**E**): Each filled point shows mean MEPP amplitude in one muscle following incubation in emGFP-865:TetC at the concentrations indicated. Black points in the control column show aggregated mean MEPP amplitudes from different muscle fibres before adding emGFP-865TetC. There was no significant effect of incubation in emGFP-865:TetC on either EPP or MEPP amplitudes; *p* > 0.20 ANOVA for both EPP and MEPP data. (**F**,**G**): Representative intracellular recordings of an EPP (blue trace) and a MEPP (inset, green trace) recorded from an isolated FDB muscle before (**F**) and after (**G**) adding emGFP-1093:TetC (50 µg/mL) to the recording chamber. (**H**,**I**): Summary data from EPP and MEPP recordings from FDB muscle fibres before and after adding emGFP-1093:TetC. Each filled blue/green symbol is the mean amplitude of EPPs/MEPPs recorded from one muscle fibre; red circles the mean values from 3 muscles and the lines indicate mean and standard deviations (SEM). There was no significant effect of emGFP-1093:TetC (*p* > 0.7, ANOVA).

**Figure 9 biomolecules-11-01499-f009:**
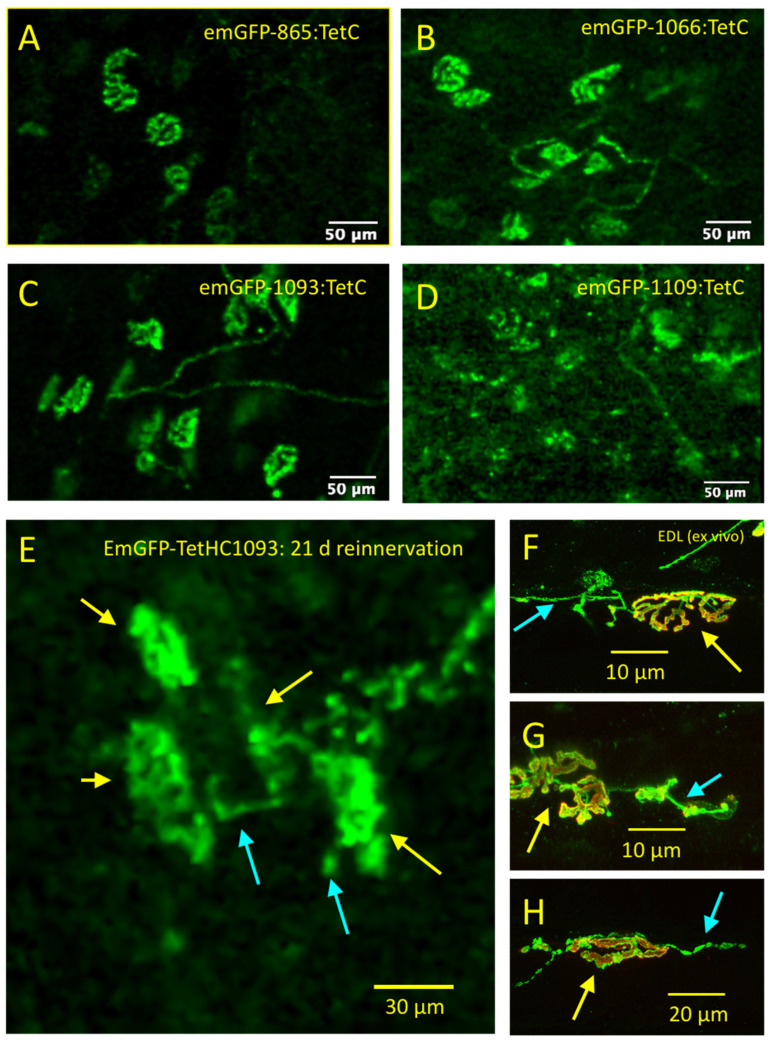
Single-waveband CEM visualizes NMJs labelled with emGFP-TetC constructs. (**A**–**D**): Still frames from videos of unfixed, vitally stained NMJs obtained by single-waveband CEM, 30–60 min after incubation in either (**A**): full length emGFP-865:TetC; (**B**): emGFP-1066:TetC; (**C**): emGFP-1093:TetC; (**D**): emGFP-1109:TetC. Staining was generally weaker and less stable with emGFP-1109:TetC compared with the other constructs. (**E**): Still frame from a single-waveband CEM showing a group of reinnervated NMJs in an isolated mouse EDL muscle, 30 days after crush of the ipsilateral sciatic nerve and staining of the isolated muscle with emGFP-1093:TetC. Some evidence of axonal sprouting (cyan arrows) is discernible (NMJs indicated by yellow arrows), (**F**–**H**): Merged images obtained by LCSM imaging of different vitally stained, unfixed NMJs in the EDL muscle, counterstained with TRITC-α-BTX, arrows indicating sprouts and NMJs, respectively, as in (**E**).

**Figure 10 biomolecules-11-01499-f010:**
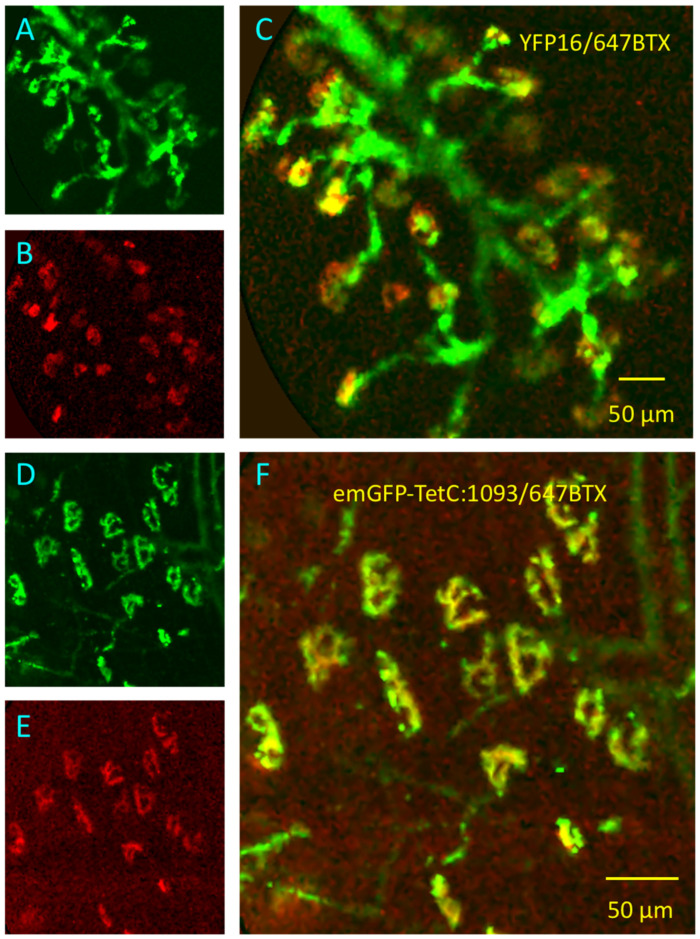
Dual-waveband CEM (DW-CEM) visualizes innervated NMJs. (**A**–**C**): Still frames from DW-CEM videos showing axons (green) and NMJs (red/yellow) in an isolated, unfixed TS muscle preparation from a *thy1.2YFP16* transgenic mouse, counterstained with Alexa647-BTX (pseudocoloured red). (**A**): YFP fluorescence; (**B**): Alexa647-BTX; (**C**): Merged images. See also Appendix A. (**D**–**F**): Still frames from DW-CEM videos of unfixed, vitally stained preparation showing motor nerve terminals after staining with emGFP-1093:TetC (**D**), Alexa647-BTX (**E**) and the merged frame (**F**). See also Appendix A.

**Figure 11 biomolecules-11-01499-f011:**
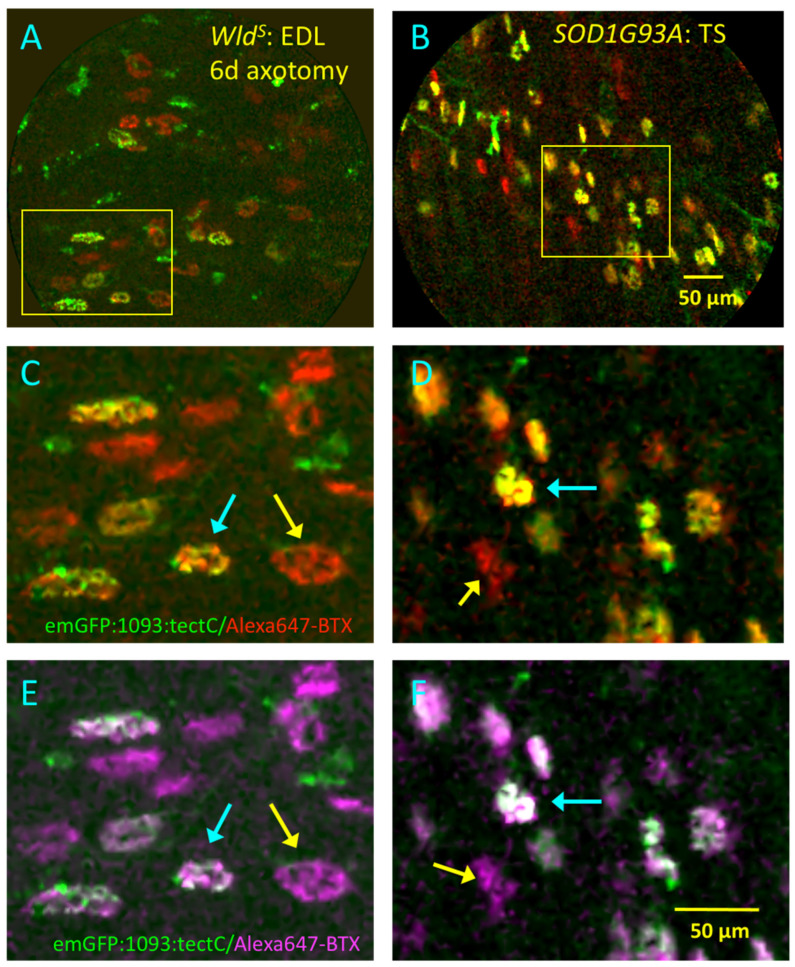
DW-CEM distinguishes innervated from denervated NMJs in mice with slow axonal degeneration. (**A**,**C**,**E**): Still frames from DW-CEM videos of unfixed, vitally stained isolated *Wld^S^* mouse EDL muscle, 6 days after sciatic nerve section, which induces slow nerve terminal and axonal degeneration in this natural mutant. (**A**): merged channels of fluorescence after incubation in emGFP-1093:TetC and Alexa647-BTX. See also Appendix A; (**C**): magnified image of the region shown boxed by the yellow rectangle in A, showing denervated (red; Alexa647-BTX staining only) and innervated (dual staining, yellow) in the same region of the axotomized muscle. (**E**): Alternative pseudocolouring (magenta for Alexa647-BTX staining; green for emGFP-1093:TetC staining; colocalization appears white) of the image shown in (**C**). (**B**,**D**,**F**): Still frames from DW-CEM videos of an isolated, unfixed and vitally stained TS muscle from a presymptomatic SOD1G93A mouse. (**B**): merged channels of fluorescence after incubation in emGFP-865:TetC and Alexa647-BTX; (**D**): magnified image of the region shown boxed by the yellow rectangle in (**B**), showing denervated (red; Alexa647-BTX staining only) and innervated (dual staining, yellow) in the same region of the diseased muscle; cyan arrows indicate instances of innervated NMJs, yellow arrows indicate examples of denervated NMJs, showing only Alexa647-BTX staining. (**F**): Alternative pseudocolouring (magenta for Alexa647-BTX staining, as in (**E**)) of the image shown in (**D**).

## Data Availability

Requests for emGFP-TetC constructs, raw data used in electrophysiological analysis, or images obtained using LSCM or CEM in the present study should be directed to the corresponding authors.

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
