# Peer review of "Confocal Endomicroscopy of Neuromuscular Junctions Stained with Physiologically Inert Protein Fragments of Tetanus Toxin"

_biomolecules, 2021, doi:10.3390/biom11101499_

Round 1

Reviewer 1 Report

In this manuscript, Roesl and colleagues study the efficacy of several truncated derivatives of the tetanus toxin C-fragment (TetC) to label presynaptic motor nerve terminal at the neuromuscular junction (NMJ). These derivatives were conjugated to GFP, with the Emerald Fluorescent Protein (emGFP), and with Alexa fluorophores. Different constructs were tested after vital staining to label the presynaptic domain of NMJs from different species, at different stages of development, and in different animal models of conditions affecting NMJ integrity. Presynaptic staining with TetC derivatives was also studied in situ or in isolated muscle preparations using fiber-optic confocal endomicroscopy (CEM). The manuscript is well written and the results are very interesting, as the combination of TetC derivatives and CEM appear as powerful tools to facilitate in vivo analyses of normal and damaged NMJs.

Although not strictly necessary for publication, I would strongly encourage the authors to consider some improvements to solidify key findings of this study. My main comments are:

. Images and videos combining live presynaptic labeling with TetC derivatives and CEM are great. Including samples from control animals examined in the exact same way will reinforce the idea that this approach has the potential to discriminate situations leading to NMJ partial or total denervation.

. Please consider improvements to the artwork of most figures (labeling of panels, font size, alignment of images, homogenize plots size and axes, and so on) as they will help to better appreciate the very nice results of this manuscript.

Other comments are:

. In Fig. 1, is it possible that samples stained with TetC derivatives can be further stained with classical presynaptic markers? (e.g. neurofilaments + synaptic proteins). This will reinforce the message that TetC derivatives specifically stain presynaptic motor terminals.

. In Fig. 2, I suggest labeling plots, as their axes are the same but they refer to different regions (which is clear only after reading the figure caption). I also suggest labeling with arrows of different colors the regions used to quantify and build the plot in panel D.

. I recommend fusing Figures 3 and 4.

. In Fig. 4, I suggest building the figures in a way that helps better to draw the conclusions. For instance, the text says: “the fluorescence was brighter with emGFP1093:TetC and emGFP-10gg:TetC compared with full-length emGFP-865:TetC” but full-length images and derivatives are depicted in different panels. Were these images acquired with similar gain and are the same z planes number? 
. In this and other figures, please clarify in the figure caption if images are vital staining or IHC post-fixation.
. Why in some of the images in Fig. 4 the axons are visible?
. Please check the image of emGFP-TetC:1093 in mouse muscle, as the image seems oversaturated.
. Please separate channels from emGFP and BTX to better compare the intensities of the different TetC derivatives. 

. In Fig. 6, including a video of the 3D rotation of panel B will help to discard that the stained axons are not motor axons.
. Is it possible to further stain these samples with an anti S100beta antibody to reinforce the idea that the ones stained with TetC derivatives are unmyelinated axons?

. In Fig. 7, please clarify what the authors call “optimal concentrations”, as the ones in the results section and the figure caption are different. Indeed, based on the first section of results, optimal concentrations were between 50-100 ug/ml, but data in Fig 7 come from experiments using 20 ug/ml.

. In Fig. 8, I would recommend including the quantification of mEPP frequency and the quantal content, as they are relevant parameters to assess presynaptic function. For instance, changes in the frequency of mEPPs could indicate an altered probability of neurotransmitter release (see PMID 11160378).

. In Fig. 9, please specify what is highlighted with yellow and cyan arrows.

. In Fig. 11, please specify the pseudo-coloring key within the figure.

Please check the text for some typos and maybe incomplete information. For instance:

. Line 35 - such “as”
. Line 203 - “fusion” instead of fusino
. Line 328 – does this sentence refer to data not shown?
. Line 521 – “translate” instead of translated

Author Response

We thank the Reviewer for their appreciative remarks about our paper and their time taken to provide an expert, detailed, thoughtful and helpful review. We have responded as best we can under the present working circumstances to all of the suggested improvements as follows:

  1. Reviewer: Images and videos combining live presynaptic labeling with TetC derivatives and CEM are great. Including samples from control animals examined in the exact same way will reinforce the idea that this approach has the potential to discriminate situations leading to NMJ partial or total denervation.
    Author Response:  Most of the images in the paper are "control" in the sense that they were obtained from unoperated, normal mice. Specifically in relation to the dual-waveband CEM imaging of muscles containing a mixture of denervated and innervated NMJs, Figure 10D-F shows images from control muscles labelled with emGFP-1093:TetC, in which 100% of the NMJs are dual stained and therefore innervated. Images from the unoperated control side of WldS mice, contralateral to the axotomised side, had a similar appearance; likewise, unaffected littermates of the SOD1G93A mice.  We feel that Figure 10 and 11 taken together, make the point satisfactorily. We have placed more emphasis on this point in the text.
  2. Reviewer: Please consider improvements to the artwork of most figures (labeling of panels, font size, alignment of images, homogenize plots size and axes, and so on) as they will help to better appreciate the very nice results of this manuscript.
    Author Response: We thank the reviewer for highlighting this. We have been through and rectified all the Figures, tidying up those with problems of alignments, font sizes etc.
  3. Reviewer: In Fig. 1, is it possible that samples stained with TetC derivatives can be further stained with classical presynaptic markers? (e.g. neurofilaments + synaptic proteins). This will reinforce the message that TetC derivatives specifically stain presynaptic motor terminals.
    Response: Unfortunately, GFP-TetC staining of nerve terminals does not endure fixation very well, so our attempts to prove co-localisation of TetC and NF/Sv2 staining were unsuccessful. We have added text to this effect. We feel the sum of evidence in the paper, taken together with other published literature on TetC binding, should give sufficient confidence that our derivatives are staining motor nerve terminals rather than some other structure. Our emGFP-TetC derivatives will become available on publication of the present paper to any researcher who may wish to   attempt this.
  4. Reviewer: In Fig. 2, I suggest labeling plots, as their axes are the same but they refer to different regions (which is clear only after reading the figure caption). I also suggest labeling with arrows of different colors the regions used to quantify and build the plot in panel D.
    Author response: The filled circels in panel C are taken from the region of the NMJ shown in panel B; open circles are from a different NMJ. The data in panel D were from regions of other NMJs that contained hotspots but we did not have sufficient resolution to bleach only hotspots; thus, hotspots were still visible after bleaching in these images. We feel that Including these images could actually be confusing so we prefer to present only the quantitative measurements. (We have adjusted the X-axes of panels C and D to be the same.) An alternative would be to remove the hot-spot FRAP data and graph from the paper. We would prefer to keep them but if the reviewer and editor feel strongly on this point we would comply.
  5. Reviewer: I recommend fusing Figures 3 and 4.
    Author response: In our working drafts of this paper, versions of Figs 3 & 4 were indeed combined. However, we were not satisfied with the appearance, which seemed cluttered, so we decided to split the figure into the present two, for clarity. We therefore think it would be better to retain Figures 3 and 4 as they are.
  6. Reviewer: In Fig. 4, I suggest building the figures in a way that helps better to draw the conclusions. For instance, the text says: “the fluorescence was brighter with emGFP1093:TetC and emGFP-10gg:TetC compared with full-length emGFP-865:TetC” but full-length images and derivatives are depicted in different panels. Were these images acquired with similar gain and are the same z planes number? 
    Author response: As noted in Results text, we did not attempt to quantify our subjective impressions regarding differences in image brightness or contrast when staining with different emGFP-TetC derivatives, so these remain subjective. The main point is that terminals stained with emGFP-865:TetC, emGFP-1066:TetC or emGFP-1093:TetC were reliably visualised with high contrast. Unfortunately, we did not carry out any experiments comparing muscles stained simultaneously with different emGFP-nnnn:TetC derivatives under identical conditions, and imaged with identical confocal settings; so our statements on this matter should be viewed with caution. We have added some extra text to this effect. The images shown were obtained on different occasions by more than one investigator and so the settings on the microscope were not the same.
  7. Reviewer: In this and other figures, please clarify in the figure caption if images are vital staining or IHC post-fixation.
    Author response: This is an important point. Our GFP-TetC derivatives are only useful as vital stains and, as indicated above, do not survive fixation very well. We have explicitly indicated in all the relevant Figure captions that the TetC staining is vital staining of unfixed tissue.
  8. Reviewer: Please check the image of emGFP-TetC:1093 in mouse muscle, as the image seems oversaturated.
    Author response: Our re-inspection of the original image shows that the Reviewer's observation is correct. Unfortunately there is nothing we can do about this post hoc, as the image was evidently saturated at acquisition (due to excess laser power). However, in mitigation,  other images in the paper showing emGFP-1093:TetC staining are not saturated and thus show more discernible detail.
  9. Reviewer: Please separate channels from emGFP and BTX to better compare the intensities of the different TetC derivatives.
    Author response: We thank the Reviewer for this isuggestion. For avoidance of doubt we have added small (grayscale) images of the green channel as insets in each of the panels shown in Figure 4C.
  10. Reviewer: In Fig. 6, including a video of the 3D rotation of panel B will help to discard that the stained axons are not motor axons.
    Author response: This is an excellent suggestion. Two supplementary videos has been added (new Supplementary Videos 1 and 2) and the original set of videos renumbered.
  11. Reviewer:  Is it possible to further stain these samples with an anti S100beta antibody to reinforce the idea that the ones stained with TetC derivatives are unmyelinated axons?
    Author response: This is another excellent suggestion. As indicated above, staining of nerve terminals with TetC derivatives normally does not survive fixation. Unfortunately, there is not sufficient time to organise and attempt a combined, superposition of vital staining with TetC and post-fixation S100 staining, given the requirement to revise the ms within time allowed for revision of the ms. So we leave this for others to attempt.
  12. Reviewer: In Fig. 7, please clarify what the authors call “optimal concentrations”, as the ones in the results section and the figure caption are different. Indeed, based on the first section of results, optimal concentrations were between 50-100 ug/ml, but data in Fig 7 come from experiments using 20 ug/ml.
    Author response: We thank the reviewer for spotting this inconsistency and we apologise for the confusion, compounded by our mistaken descriptions in the legend and text. We have replaced Figure 7 with the recording we had intended to show here (making the same point), and which was from a muscle before and after incubation in 50 µg/ml emGFP-1093:TetC. The legend and text have been corrected accordingly.
  13. Reviewer: In Fig. 8, I would recommend including the quantification of mEPP frequency and the quantal content, as they are relevant parameters to assess presynaptic function. For instance, changes in the frequency of mEPPs could indicate an altered probability of neurotransmitter release (see PMID 11160378).
    Author response: The Reviewer is absolutely correct. We have now added MEPP frequency data (in the text). This was inexplicably omitted from the previous version. There was a small apparent increase in mean MEPP frequency after adding emGFP-TetC but the difference was not statistically significant (P=0.243, paired t-test; N=4 muscles). An accurate quantal analysis was beyond the scope of the study as it would have required voltage clamp analysis of evoked EPCs and spontaneous MEPCs (due to non-linear summation of synaptic potentials at the levels of release recorded here). However, the absence of effect of emGFP-TetC variants on either EPP or MEPP amplitude suggests, implicitly, that quantal content is unlikely to have been significantly altered and we have now stated this in the text.
  14. Reviewer: In Fig. 9, please specify what is highlighted with yellow and cyan arrows.
    Author response: Our apologies for this oversight. The indications of the yellow and cyan arrows has now been inserted in the legend to Fig 9.
  15. Reviewer: In Fig. 11, please specify the pseudo-coloring key within the figure.
    Author response: Apologies for neglecting this: corrected.
  16. Reviewer: Please check the text for some typos and maybe incomplete information.
    Author response: We apologise for our sub-standard proof reading. We believe we have now corrected all typos and spelling errors.

Reviewer 2 Report

Comments and Suggestions for Authors

Confocal laser endomicroscopy is endoscopic technology that was originally proposed for the permeation high-resolution assessment of gastrointestinal mucosal histology at a cellular and sub-cellular level. At the last time, it is considered as the instrument for vital imaging of the neurological structures. The authors convincingly show that fragments of tetanus toxin chains conjugated to a fluorescent protein are a good marker for neuromuscular contacts.

A few remarks

1) It needs to show more clearly justified the benefits of the fiber-optic confocal endomicroscopy in comparisons of the laser scanning confocal microscopy for neuromuscular junction imaging.

2) How can the authors explain the binding of tetanus toxin fragments to sites in the extra synaptic zone (since they demonstrate that “in some preparations, emGFP-Tet constructs produced notable, additional labeling of sparsely-distributed axons, many associated with blood vessels”) and how, in this case, to separate the axons that form the synapse from the degenerated or related to the autonomic nerves.

3) Practically nothing is said about the sites of the axon membrane that bind the modified toxin, except that it is ganglioside and peptide binding sites. It is necessary to explain exactly what are these sites of the neuronal membrane are and why they can move over time.

Author Response

We are very grateful for the Reviewer's appreciative remarks about our paper and we thank the Reviewer for their time and helpful comments in reviewing it. We have endeavoured to incorporate responses to the Reviewer's suggestions in the body of the paper, as follows:

  1. Reviewer: It needs to show more clearly justified the benefits of the fiber-optic confocal endomicroscopy in comparisons of the laser scanning confocal microscopy for neuromuscular junction imaging.
    Author response: The principal advantages of fibre-optic CEM are capability for in situ, real time imaging of fluorescent structures, using a flexible, hand-held miniature probe. Potentially, CEM enables minimally invasive live imaging of NMJs in any muscle in situ via a small (<0.5 cm) skin incision, as we have shown in our previous studies using mice. This is not possible using conventional LSCM. We have added the above text to the relevant section of the Discussion.
  2. Reviewer: How can the authors explain the binding of tetanus toxin fragments to sites in the extra synaptic zone (since they demonstrate that “in some preparations, emGFP-Tet constructs produced notable, additional labeling of sparsely-distributed axons, many associated with blood vessels”) and how, in this case, to separate the axons that form the synapse from the degenerated or related to the autonomic nerves.
    Author response: These are excellent questions to which unfortunately we do not yet have clear answers or biological explanations. In most adult preparations, our emGFP-TetC derivatives selectively stain motor nerve terminals only, although motor axons were evidently stained as well in neonatal preparations (see Figure 5). In a minority of adult preparations, short lengths of preterminal axons are also stained. In additon there was sometimes staining of axons associated with blood vessels, or staining of small - we presume unmyelinated - sensory or autonomic axons distributed throughout the muscle and that were not associated with NMJs (see response to Revieer 1 and new Supplementary Videos 1 and 2). We have added a brief paragraph to the Discussion on this issue.
  3. Reviewer: Practically nothing is said about the sites of the axon membrane that bind the modified toxin, except that it is ganglioside and peptide binding sites. It is necessary to explain exactly what are these sites of the neuronal membrane are and why they can move over time.
    Author response: We agree with the Reviewer that it will be important to establish what, exactly, are the components of the nerve terminal membrane to which tetanus toxin binds via the binding site on its C-fragment. However, since we did not perform any experiments on this, any discussion - beyond recapitulating what is known about the peptide or ganglioside receptors referred to - would be overly speculative on our part. So we prefer to avoid this and leave to others the resolution of this important issue via appropriate experimental investigation.

Reviewer 3 Report

An interesting study exploring a potentially very significant technical development for imaging of living neuromuscular junctions (NMJs) in rodents and humans. The development of a range of truncated and ostensibly non-toxic fluorescently tagged TetC-chain of tetanus toxin of various lengths is shown to label NMJs clearly and robustly, in healthy, diseased and neonatal muscles. The demonstration of minimal or no effect of labelling on NMJ functionality for the duration of the experiments is convincing. Overall, a convincing and well-constructed study.

I have only a few minor comments - some specific and some general.

General Comments:

1) The ms does not seem to give a substantial rationale for why a range of lengths of TetC were tested. It would be useful for this to be explained throughout.

2) The conclusion seems to gloss over some of the issues that are quite rightly identified in the main text. In particular, the inability to image human NMJs by CEM in situ implies that the technique is currently restricted to either rodent models or human muscle explants. The authors state, 'The fluorescence intensity of these conjugates are sufficient for visualisation of NMJs in situ, using confocal endomicroscopy (CEM).' The current wording thus seems too sweeping in its generalisation of the technique's undoubted utility, by failing to acknowledge this significant technical limitation. 

3) While the evidence for a lack of acute toxicity is very clear and convincing, this does not mean longer-term presence of the toxin at NMJs (days to weeks) is non-toxic. It is possible that the toxin might be internalised and transported to the soma, and potentially have effects there. The presence of granular inclusions in labeled terminals in Figure 1 might indicate this is happening. A statement acknowledging that such studies are needed, or that were not performed here, would be useful.

Minor issues:

Several typographical errors throughout - wrong tense, NMJ's instead of NMJs for plural, capitalisation of Video, etc. Inconsistent use of bold or not bold style for the word 'Figure' throughout the text. And, for the letters in the legend for figure sections (e.g. Figure 1 legend, none of the section letters are bold, whereas they usually are for most other figure legends).

It is not clear why the word 'video' is often capitalised in the text. Presumably this should always be lower case, except when used for 'Supplementary Video'?

Methods - experiments on human tissue were approved by an ethical committee. I think a similar ethical statement is required for the animal experiments. Since the animals were anaesthetised for at least part of many protocols, these would be regulated procedures under ASPA. There is a statement the animals were housed according to HO regulations, but not that the procedures had received ethical approval.

Methods - was the MPS gassed with air, as stated? It is more usual to gas with 95% O2/5% CO2.

Methods - Sylgard supplier?

Line 315-7 - citation needed that construct excluded A and B regions.

Line 326 -7 - explain '.. the location and short lengths of the preterminal axon ..' What is meant by 'location'?

Line 331-2 - explain how a mean recovery involves stating a range (64-69%), and that is derived from on an n=2. A mean should be a single number (+/- range/SD/SE). Can an 'n of 2' be used to generate a meaningful mean?

L342 - why was this modification performed? As per general comments above - what was the rationale for modifying GFP-TetC. Why was it of interest or would it be a useful procedure? Similarly L351 - why was the truncation necessary/potentially useful?

Line 753 - what therapeutics might be usefully delivered and for which disease(s)?

Author Response

We thank the Reviewer for their complimentary remarks on the interest of our paper and for their time in carefully reading and reviewing the manuscript. Our responses to the Reviewer's comments and suggestions are as follows:

  1. Reviewer: The ms does not seem to give a substantial rationale for why a range of lengths of TetC were tested. It would be useful for this to be explained throughout.
    Author Response: Apologies if this was not as clear as it could have been. We previously stated in the Introduction "Remarkably, truncations of TetC from the N-terminus were shown previously to improve binding to ganglioside receptors." and cited reference 30 (Halpern & Loftus, 1993). To make this clearer, we have added "therefore" to the sentence that follows it in the Introduction; and we have added an explanatory sentence to the opening paragraph of the Discussion to re-emphasise this point.
  2. Reviewer: The conclusion seems to gloss over some of the issues that are quite rightly identified in the main text. In particular, the inability to image human NMJs by CEM in situ implies that the technique is currently restricted to either rodent models or human muscle explants. The authors state, 'The fluorescence intensity of these conjugates are sufficient for visualisation of NMJs in situ, using confocal endomicroscopy (CEM).' The current wording thus seems too sweeping in its generalisation of the technique's undoubted utility, by failing to acknowledge this significant technical limitation. 
    Author response: We intended our conclusion to end on an "upbeat" but the Reviewer is right to draw attention to the shortcomings of CEM/TetC labelling with respect to potential future clinical applications so we have now made this clear in the Conclusion.
  3. Reviewer: While the evidence for a lack of acute toxicity is very clear and convincing, this does not mean longer-term presence of the toxin at NMJs (days to weeks) is non-toxic. It is possible that the toxin might be internalised and transported to the soma, and potentially have effects there. The presence of granular inclusions in labeled terminals in Figure 1 might indicate this is happening. A statement acknowledging that such studies are needed, or that were not performed here, would be useful.
    Author response: We agree this will be important to establish. We have added a paragraph to this effect to the Discussion, at the end of section 4.1
  4. Reviewer: Several typographical errors throughout - wrong tense, NMJ's instead of NMJs for plural, capitalisation of Video, etc. Inconsistent use of bold or not bold style for the word 'Figure' throughout the text. And, for the letters in the legend for figure sections (e.g. Figure 1 legend, none of the section letters are bold, whereas they usually are for most other figure legends). It is not clear why the word 'video' is often capitalised in the text. Presumably this should always be lower case, except when used for 'Supplementary Video'?
    Author response: We apologise for these multiple errors and inconsistencies. We have spell-checked the revised ms and corrected as many of the other errors as we have found. "Figure" has been changed to lower case throughout, and "video" substituted for 'Video' except when referring explicitly to "Supplementary Video".
  5. Reviewer: Methods - experiments on human tissue were approved by an ethical committee. I think a similar ethical statement is required for the animal experiments. Since the animals were anaesthetised for at least part of many protocols, these would be regulated procedures under ASPA. There is a statement the animals were housed according to HO regulations, but not that the procedures had received ethical approval.
    Author response: We thank the Reviewer for drawing attention to this important omission. We have now indicated in Methods that the surgical denervations in mice were also conducted with local ethical approval and under Home Office Licence.
  6. Reviewer: Methods - was the MPS gassed with air, as stated? It is more usual to gas with 95% O2/5% CO2.
    Author response: The text is correct. We routinely use air to bubble our HEPES-buffered mammalian physiological saline (MPS). 95%O2/5%CO2 bubbling is required only when using bicarbonate-buffered physiological salines (such as Liley's or Kreb's solutions). Some labs bubble HEPES-buffered salines with 100% O2. However, we found several years ago in controlled experiments that bubbling HEPES-MPS with air (using an aquarium pump rather than cumbersome gas cyclinders) was sufficient for conducting standard electrophysiological or muscle force measurements at room temperature.
  7. Reviewer: Methods - Sylgard supplier?
    Author response: Now indicated (Dow).
  8. Reviewer: Line 315-7 - citation needed that construct excluded A and B regions.
    Author response: This was covered by citation of reference 30 in the preceding sentence but we have now moved this citation to the end of the  sentence that the Reviewer refers to.
  9. Reviewer: Line 326 -7 - explain '.. the location and short lengths of the preterminal axon ..' What is meant by 'location'?
    Author response:  "the location of" now deleted. (The description that short lengths of the preterminal axon were sometimes visible is sufficient.)
  10. Reviewer: Line 331-2 - explain how a mean recovery involves stating a range (64-69%), and that is derived from on an n=2. A mean should be a single number (+/- range/SD/SE). Can an 'n of 2' be used to generate a meaningful mean?
    Author response: A fair point. The simplest way of dealing with the offending parenthetical summary of the data is to delete it, which we have done.
  11. Reviewer: L342 - why was this modification performed? As per general comments above - what was the rationale for modifying GFP-TetC. Why was it of interest or would it be a useful procedure? Similarly L351 - why was the truncation necessary/potentially useful?
    Author response: See response to Point 1, above.
  12. Reviewer: Line 753 - what therapeutics might be usefully delivered and for which disease(s)?
    Author response: This is not an original idea, as indicated by other publications including references 26 and 69-71 (cited) but we felt it was one worth re-emphasising. Reference 70, for example, shows that GDNF can be delivered in this fashion. We have added ".. as novel treatments for ALS or other neuromuscular diseases" to the relevant sentence in the Discussion.